# HyperMARL: Adaptive Hypernetworks for Multi-Agent RL

**Kale-ab Abebe Tessera[1]**    **Arrasy Rahman[2]**    **Amos Storkey[1]**    **Stefano V. Albrecht[3]**

[1]School of Informatics, University of Edinburgh, Edinburgh, UK
[2]School of Computer Science, University of Texas at Austin, Austin, TX, USA
[3]DeepFlow, London, UK
{k.tessera,a.storkey}@ed.ac.uk, arrasy@utexas.edu

## Abstract

Adaptive cooperation in multi-agent reinforcement learning (MARL) requires policies to express homogeneous, specialised, or mixed behaviours, yet achieving this adaptivity remains a critical challenge. While parameter sharing (PS) is standard for efficient learning, it notoriously suppresses the behavioural diversity required for specialisation. This failure is largely due to cross-agent gradient interference, a problem we find is surprisingly exacerbated by the common practice of *coupling agent IDs with observations.* Existing remedies typically add complexity through altered objectives, manual preset diversity levels, or sequential updates – raising a fundamental question: *can shared policies adapt without these intricacies?* We propose a solution built on a key insight: an agent-conditioned hypernetwork can generate agent-specific parameters and *decouple* observation- and agent-conditioned gradients, directly countering the interference from coupling agent IDs with observations. Our resulting method, *HyperMARL*, avoids the complexities of prior work and empirically reduces policy gradient variance. Across diverse MARL benchmarks (22 scenarios, up to 30 agents), HyperMARL achieves performance competitive with six key baselines while preserving behavioural diversity comparable to non-parameter sharing methods, establishing it as a versatile and principled approach for adaptive MARL. The code is publicly available at https://github.com/KaleabTessera/HyperMARL.

## 1 Introduction

Specialist and generalist behaviours are critical to collective intelligence, enhancing performance and adaptability in both natural and artificial systems [52, 43, 44, 23, 50]. In Multi-Agent Reinforcement Learning (MARL) [2], this translates to a critical need for policies that can flexibly exhibit specialised, homogeneous, or mixed behaviours to meet diverse task demands [26, 6].

Optimal MARL performance thus hinges on being able to represent the required behaviours. While No Parameter Sharing (NoPS) [30] enables specialisation by using distinct per-agent networks, it suffers from significant computational overhead and sample inefficiency [11]. Conversely, Full Parameter Sharing (FuPS) [45, 18, 15], which trains a single shared network, improves efficiency but typically struggles to foster the behavioural diversity necessary for many complex tasks [24, 17, 26].

This failure of FuPS, particularly for diverse behaviours, was hypothesised to be gradient interference among agents, whereby their updates negatively impact each other's learning [11, 54]. We not only empirically validate this hypothesis but also demonstrate a critical insight: this conflict is significantly exacerbated by the common practice of *coupling observations with agent IDs* within a shared network (Fig. 1 for coupling, Sec. 3.2 for results).

39th Conference on Neural Information Processing Systems (NeurIPS 2025).

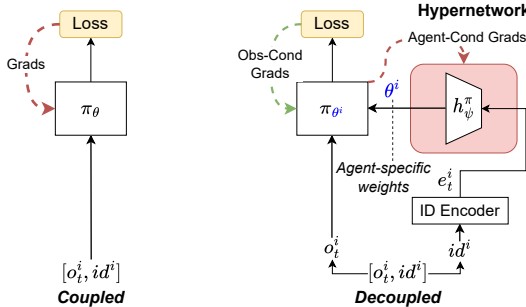

Figure 1: *HyperMARL Policy Architecture.* Common agent-ID conditioned shared MARL policy (FuPS+ID, left) vs HyperMARL (right), which uses an agent-conditioned hypernetwork to generate agent-specific weights and *decouples* observation- and agent-conditioned gradients.

Balancing FuPS efficiency with the capacity for diverse behaviours therefore remains a central open problem in MARL. Prior works have explored intrinsic-rewards [26, 22], role-based allocations [48, 49], specialised architectures [24, 28, 6], sequential updates [54], and sharing parameters within clusters of agents [11]. However, these remedies introduce their own intricacies: they often alter the learning objective, require prior knowledge of optimal diversity levels, necessitate maintaining agent-specific parameters or require sequential updates. This raises a fundamental question: *Can we design a shared MARL architecture that flexibly supports both specialised and homogeneous behaviours—without altered learning objectives, manual preset diversity levels, or sequential updates?*

Guided by our observation-ID coupling insight, we propose *HyperMARL*, a novel agent-conditioned hypernetwork [19] architecture. HyperMARL generates per-agent weights on the fly (Fig. 1) and explicitly *decouples* observation- and agent-conditioned gradients (Sec. 4.3). This choice is motivated by hypernetworks' proven effectiveness at resolving gradient conflicts in multi-task RL [33] and continual learning [47]. Our work establishes their effectiveness for the problem of cross-agent interference in MARL. Indeed, HyperMARL empirically attains lower policy gradient variance than FuPS, and we show this decoupling is critical for specialisation (Sec. 5.2, 6.1), confirming its role in mitigating interference.

We validate HyperMARL on diverse MARL benchmarks – including Dispersion and Navigation (VMAS) [5], Multi-Agent MuJoCo (MAMuJoCo) [37], SMAX [41], and Blind-Particle Spread (BPS) [11] – across environments with two to thirty agents that require homogeneous, heterogeneous, or mixed behaviours. HyperMARL consistently matches or outperforms NoPS, FuPS, and diversity-promoting methods such as Diversity Control (DiCo) [6], Heterogeneous-Agent Proximal Policy Optimisation (HAPPO) [54], Kaleidoscope [28] and Selective Parameter Sharing (SePS) [11], while achieving NoPS-level behavioural diversity while using a shared architecture.

Our contributions are as follows:

- We identify that cross-agent gradient interference in shared policies is critically exacerbated by the common practice of coupling agent IDs with observations (Sec. 3.2).

- We propose *HyperMARL* (Sec. 4), an agent-conditioned hypernetwork architecture, to test the hypothesis that explicitly *decoupling* these gradients enables adaptive (diverse, homogeneous, or mixed) behaviours without the complexities of prior remedies (e.g., altered objectives, preset diversity levels, or sequential updates).

- Our extensive evaluation (Sec. 5) across 22 diverse scenarios (up to 30 agents) shows HyperMARL achieves competitive returns against six strong baselines, while achieving NoPS-level behavioural diversity. We further show this decoupling is empirically linked to reduced policy gradient variance and is critical for specialisation (Sec. 5.2; Sec. 6.1).

## 2   Background

We formulate the fully cooperative multi-agent systems addressed in our work as a Dec-POMDP [34]. A Dec-POMDP is a tuple, $\langle \mathbb{I}, \mathbb{S}, \{\mathbb{A}^i\}_{i \in \mathbb{I}}, R, \{\mathbb{O}^i\}_{i \in \mathbb{I}}, O, T, \rho_0, \gamma \rangle$, where $\mathbb{I}$ is the set of agents of size

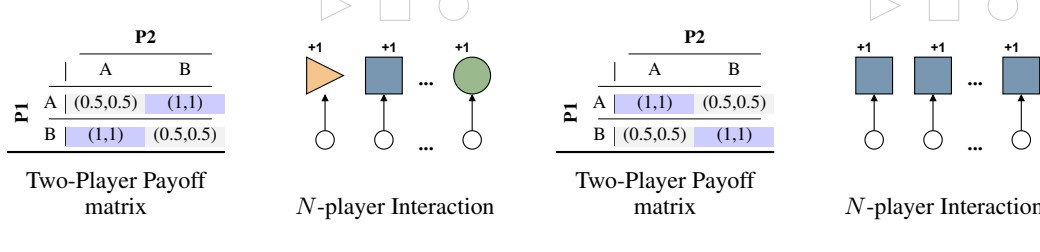

|       | **(a)** Specialisation Game | **(b)** Synchronisation Game |
|-------|----------------------------|------------------------------|

Figure 2: *Specialisation and Synchronisation Games.* The Specialisation game (left), which encourages *distinct* actions, and the Synchronisation game (right), where rewards encourage *identical* actions. Depicted are their two-player payoff matrices (pure Nash equilibria in blue) and $N$-player interaction schematics. While simple in form, these games are challenging MARL benchmarks due to non-stationarity and exponentially scaling observation spaces (temporal version).

$n = |\mathbb{I}|$, $\mathbb{S}$ is the set of global states with an initial state distribution $\rho_0$, $\mathbb{A}^i$ is the action space for agent $i$ where $\mathbb{A} = \times_i \mathbb{A}^i$ is the joint action space, $R : \mathbb{S} \times \mathbb{A} \to \mathbb{R}$ is the shared reward function, $\mathbb{O}^i$ is the observation space for agent $i$ with the joint observation space $\mathbb{O} = \times_i \mathbb{O}^i$, $O : \mathbb{O} \times \mathbb{A} \times \mathbb{S} \to [0, 1]$ is the probability of joint observation $\mathbf{o} \in \mathbb{O}$, i.e. $O(\mathbf{o}, \mathbf{a}, s) = \Pr(\mathbf{o_t}|s_t, \mathbf{a_{t-1}})$, $T : \mathbb{S} \times \mathbb{A} \times \mathbb{S} \to [0, 1]$ is the state transition function i.e. $T(s, \mathbf{a}, s') = \Pr(s_{t+1} \mid s_t, \mathbf{a}_t)$ and $\gamma$ is the discount factor.

In this setting, each agent $i$ receives a partial observation $o_t^i \in \mathbb{O}^i$. These observations are accumulated into an action-observation history $h_t^i = (o_0^i, a_0^i, \ldots, o_{t-1}^i, a_{t-1}^i, o_t^i)$. Each agent $i$ acts based on their decentralised policies $\pi^i(a^i|h^i)$. The joint history and joint action are defined as follows $\mathbf{h_t} = (h_t^1, \ldots, h_t^n)$ and $\mathbf{a_t} = (a_t^1, \ldots, a_t^n)$. The goal is to learn an optimal joint policy $\boldsymbol{\pi^*} = (\pi^{1*}, \ldots, \pi^{n*})$ that maximizes the expected discounted return as follows,[1] $\boldsymbol{\pi^*} = \arg\max_{\boldsymbol{\pi}} \mathbb{E}_{s_0 \sim \rho_0, \, \mathbf{h} \sim \boldsymbol{\pi}} [G(\mathbf{h})]$, where $G(\mathbf{h}) = \sum_{t=0}^{\infty} \gamma^t R(s_t, \mathbf{a}_t)$.

**Specialised Policies and Environments.** We say an environment is *specialised* if its optimal joint policy contains at least two distinct, non-interchangeable agent policies ( Def. 1 in App. C). Under this mild condition, tasks such as Dispersion (5.2) or our Specialisation Game (E.1) require agents to learn complementary roles rather than identical behaviours.

## 3 Are Independent or Fully Shared Policies Enough?

Standard independent (**NoPS**) and fully parameter-shared (**FuPS**) policies face inherent trade-offs in MARL. NoPS allows for uninhibited agent specialisation but can be sample inefficient and computationally expensive. FuPS, often conditioned with an agent ID (FuPS+ID), is more efficient but can struggle when agents must learn diverse behaviours [11]. This section investigates the limitations of these common policy architectures.

To probe these limitations, we introduce two illustrative environments: the **Specialisation Game**, rewarding *distinct* actions, and the **Synchronisation Game**, rewarding *identical* actions. Both are inspired by prior work [10, 17, 5, 35] and extended here to $N$-agent and temporal settings where agents observe prior joint actions (see Appendix E for full definitions).

### 3.1 Limitations of Fully Shared and Independent Policies

FuPS *without* agent IDs provably cannot recover optimal pure Nash equilibria in the non-temporal 2-player Specialisation Game (Proof E.3, App. E). In practice, however, FuPS is often conditioned *with* agent IDs, and MARL policies must handle complexities beyond static, two-player interactions. We therefore evaluate standard architectures in the temporal $n$-player versions of these games[2] We compare three standard architectures trained with REINFORCE [51]: 1) **NoPS**: independent

---

[1] We use simplified notation here omitting explicit dependence on state transitions and distributions for brevity.

[2] Results for non-temporal (normal-form) variants are in App. E.5.

Table 1: Average evaluation reward (mean $\pm$ 95% CI) for *temporal* Specialisation vs. Synchronisation using REINFORCE (10 seeds). **Bold**: highest mean, no CI overlap. Neither fully shared nor independent policies consistently achieve the highest mean reward.

| #Ag | Specialisation | | | Synchronisation | | |
|---|---|---|---|---|---|---|
| | NoPS | FuPS | FuPS+ID | NoPS | FuPS | FuPS+ID |
| 2 | **0.88**±0.09 | 0.50±0.00 | 0.64±0.10 | 0.83±0.12 | **1.00**±0.00 | 0.91±0.09 |
| 4 | **0.74**±0.08 | 0.25±0.00 | 0.40±0.07 | 0.32±0.03 | **1.00**±0.00 | 0.67±0.15 |
| 8 | **0.68**±0.02 | 0.12±0.00 | 0.25±0.03 | 0.14±0.00 | **1.00**±0.00 | 0.54±0.10 |
| 16 | **0.64**±0.01 | 0.06±0.00 | 0.13±0.02 | 0.07±0.00 | **1.00**±0.00 | 0.55±0.14 |

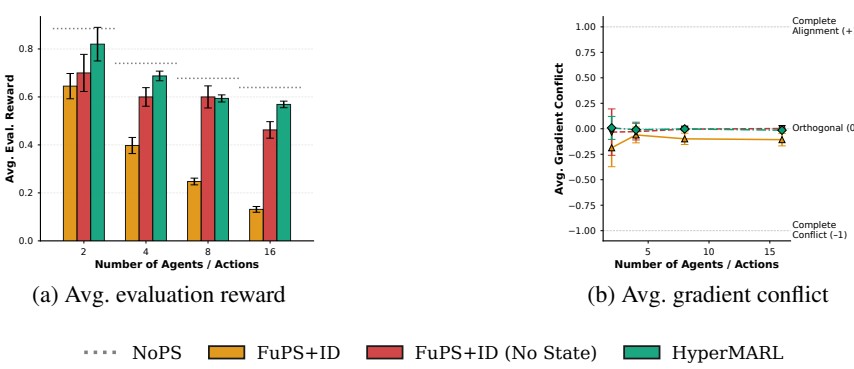

(a) Avg. evaluation reward        (b) Avg. gradient conflict

···· NoPS    ▬ FuPS+ID    ▬ FuPS+ID (No State)    ▬ HyperMARL

Figure 3: *Multi-agent policy gradient methods in the Specialisation environment.* The FuPS+ID (No State) ablation outperforms FuPS+ID, showing near-orthogonal gradients (b), indicating that observation–ID decoupling is important. HyperMARL (MLP) enables this decoupling while leveraging state information, and achieves better performance and reduced gradient conflict than FuPS+ID.

policies ($\pi_{\theta^i}(a^i|o^i)$); 2) **FuPS**: a single shared policy ($\pi_\theta(a^i|o^i)$); and 3) **FuPS+ID**: a shared policy incorporating a one-hot agent ID ($\pi_\theta(a^i|o^i, id^i)$). All use single-layer networks, 10-step episodes and $10,000$ training steps (further details in Table 9 in App. I).

**Empirical Performance.** Table 1 shows that neither NoPS nor FuPS consistently achieves the highest mean evaluation rewards. NoPS excels in the Specialisation Game but is outperformed by FuPS (optimal) and FuPS+ID in the Synchronisation Game. Furthermore, the performance gaps widen as the number of agents increases (notably at $n = 8$ and $n = 16$), highlighting the scalability challenges of both fully independent and fully shared policies.

### 3.2 Why FuPS+ID Fails to Specialise: The Problem of Gradient Conflict

Despite being a universal approximator [20], FuPS+ID often struggles to learn diverse policies in practice (Table 1, [11, 54]). A key reason is *gradient conflict*: when a single network processes both observation $o$ and agent ID $id^i$, updates intended to specialise agent $i$ (based on $id^i$) can conflict with updates for agent $j$ (based on $id^j$), particularly if they share similar observations but require different actions. This obstructs the emergence of specialised behaviours (conflict measured via inter-agent gradient cosine similarity, App. E.4).

**Importance of Observation and ID Decoupling.** To investigate the effect of entangled observation and ID inputs, we introduce an ablation: *FuPS+ID (No State)*, where the policy $\pi_\theta(a^i \mid id^i)$ conditions *only* on the agent ID, ignoring observations. Surprisingly, *FuPS+ID (No State) outperforms standard FuPS+ID* in the Specialisation Game for all tested $N$ (Figure 3a), even when $N \leq 4$ (where observation spaces are small, suggesting the issue is not merely observation size). Figure 3b reveals why: FuPS+ID (No State) shows near-zero gradient conflict (nearly orthogonal gradients), whereas standard FuPS+ID exhibits negative cosine similarities (conflicting gradients).

These results show that naively coupling observation and ID inputs in shared networks can lead to destructive interference, hindering specialisation. While discarding observations is not a general solution (most tasks require state information), this finding motivates designing architectures that can leverage both state and agent IDs, while minimising interference. Section 4 introduces HyperMARL

(Figure 1), which explicitly *decouples observation- and agent-conditioned* gradients through agent-conditioned hypernetworks, leading to improved performance over FuPS variants and reduced gradient conflict compared to standard FuPS+ID (Figure 3).

# 4 HyperMARL

We introduce *HyperMARL*, an approach that uses agent-conditioned hypernetworks to learn diverse or homogeneous policies *end-to-end*, without modifying the standard RL objective or requiring manual preset diversity levels. By operating within a fully shared paradigm, HyperMARL leverages shared gradient information while enabling specialisation through the decoupling of observation- and agent-conditioned gradients. We present the pseudocode in Sec. F.1, with additional scaling (F.3) and runtime (F.4) details.

## 4.1 Hypernetworks for MARL

As illustrated in Figure 1, for any agent $i$ with context $e^i$ (i.e., either a one-hot encoded ID or a learned embedding), the hypernetworks generate the agent-specific parameters:

$$\theta^i = h_\psi^\pi(e^i), \quad \phi^i = h_\varphi^V(e^i), \tag{1}$$

where $h_\psi^\pi$ and $h_\varphi^V$ are the hypernetworks for the policy and critic, respectively. The parameters $\theta^i$ and $\phi^i$ define each agent's policy $\pi_{\theta^i}$ and critic $V_{\phi^i}$, dynamically enabling either specialised or homogeneous behaviours as required by the task.

**Linear Hypernetworks** Given a one-hot agent ID, $\mathbb{1}^i \in \mathbb{R}^{1 \times n}$, a linear hypernetwork $h_\psi^\pi$ generates agent-specific parameters $\theta^i$ as follows[3]:

$$\theta^i = h_\psi^\pi(\mathbb{1}^i) = \mathbb{1}^i \cdot W + b \tag{2}$$

where $W \in \mathbb{R}^{n \times m}$ is the weight matrix (with $m$ the per-agent parameter dimensionality and $n$ is the number of agents), and $b \in \mathbb{R}^{1 \times m}$ is the bias vector. Since $\mathbb{1}^i$ is one-hot encoded, each $\theta^i$ corresponds to a specific row of $W$ plus the shared bias $b$. If there is no shared bias term, this effectively replicates training of separate policies for each task (in our case, for each agent) [3], since there are no shared parameters and gradient updates are independent.

**MLP Hypernetworks for Expressiveness** To enhance expressiveness, MLP Hypernetworks incorporate hidden layers and non-linearities:

$$\theta^i = h_\psi^\pi(e^i) = f_{\psi_1}^\pi \big( g_{\psi_2}^\pi(e^i) \big) \tag{3}$$

where $g_{\psi_2}^\pi$ is an MLP processing the agent context $e^i$, and $f_{\psi_1}^\pi$ is a final linear output layer.

Unlike linear hypernetworks with one-hot agent IDs, MLP hypernetworks do not guarantee distinct weights for each agent. Additionally, they increase the total number of trainable parameters, necessitating a careful balance between expressiveness and computational overhead.

## 4.2 Agent Embeddings and Initialisation

The agent embedding $e^i$ is a one-hot encoded ID for Linear Hypernetworks. For MLP Hypernetworks, we use learned agent embeddings, orthogonally initialised and optimised end-to-end with the hypernetwork. HyperMARL's hypernetworks are themselves initialised such that the generated agent-specific parameters $(\theta^i, \phi^i)$ initially match the distribution of standard direct initialisation schemes (e.g., orthogonal for PPO, preserving fan in/out), promoting stable learning.

## 4.3 Gradient Decoupling in HyperMARL

A core difficulty in FuPS is cross-agent gradient interference [11, 54]. HyperMARL mitigates this by generating each agent's parameters through a shared hypernetwork, thereby *decoupling agent-conditioned* and *observation-conditioned* components of the gradient.

---

[3]For conciseness we only show the policy parameters in this section.

**Hypernetwork gradients.** Consider a fully cooperative MARL setting with a centralised critic, we can formulate the policy gradient for agent $i$ as follows [2, 25]:

$$\nabla_{\theta^i} J(\theta^i) \ = \ \mathbb{E}_{\mathbf{h_t}, \mathbf{a_t} \sim \boldsymbol{\pi}} \left[ A(\mathbf{h_t}, \mathbf{a_t}) \, \nabla_{\theta^i} \log \pi_{\theta^i}(a_t^i \mid h_t^i) \right],$$

where $\mathbf{h_t}$ and $\mathbf{a_t}$ are the joint histories and joint actions for all agents, $\theta^i$ denotes the parameters of agent $i$, and $A(\mathbf{h_t}, \mathbf{a_t}) = Q(\mathbf{h_t}, \mathbf{a_t}) - V(\mathbf{h_t})$ is the advantage function.

**Decoupling.** In HyperMARL each agent's policy weights are produced by the hypernetwork $h_\psi^\pi$: $\theta^i = h_\psi^\pi(e^i)$, so we optimise a *single* parameter vector $\psi$. Applying the chain rule and re-ordering the expectations:

$$\nabla_\psi J(\psi) = \sum_{i=1}^{I} \underbrace{\nabla_\psi h_\psi^\pi(e^i)}_{\mathbf{J}_i \text{ (agent-conditioned)}} \underbrace{\mathbb{E}_{\mathbf{h}_t, \mathbf{a}_t \sim \boldsymbol{\pi}} \left[ A(\mathbf{h_t}, \mathbf{a_t}) \nabla_{\theta^i} \log \pi_{\theta^i}(a_t^i \mid h_t^i) \right]}_{Z_i \text{ (observation-conditioned)}}. \tag{4}$$

- **Agent-conditioned factor $\mathbf{J}_i$.** This Jacobian depends only on the fixed embedding $e^i$ and the hypernetwork weights $\psi$, therefore, it is *deterministic* with respect to mini-batch samples (as $e^i$ and $\psi$ are fixed per gradient step), separating agent identity from trajectory noise.

- **Observation-conditioned factor $Z_i$.** The expectation averages trajectory noise *per agent $i$* for its policy component $\pi_{\theta^i}$, prior to transformation by $\mathbf{J}_i$ and aggregation.

The crucial structural decoupling in Equation (4) ensures HyperMARL first averages noise per agent (via factor $Z_i$) before applying the deterministic agent-conditioned transformation $\mathbf{J}_i$. This mitigates gradient interference common in FuPS+ID [11, 54], where observation noise and agent identity become entangled (see Equation (12) in App. F.2). This is the MARL analogue of the task/state decomposition studied by [42, Eq. 18] in Meta-RL. Section 5.2 empirically verifies the predicted variance drop, and ablations in Section 6.1 demonstrate that disabling decoupling degrades performance, underscoring its critical role.

## 5 Experiments

Our empirical evaluation of HyperMARL assesses whether agent-conditioned hypernetworks can enable adaptive (specialised or homogeneous) policies without altered RL objectives, preset diversity levels, or sequential updates. We structure our experiments to directly answer two key research questions: **Q1: Specialised Policy Learning:** Can *HyperMARL* effectively learn *specialised policies* via a shared hypernetwork? **Q2: Effectiveness in Homogeneous Tasks**: Is *HyperMARL* competitive in environments that necessitate homogeneous behaviours?

To address these questions, HyperMARL is evaluated against six representative modern baselines across a carefully selected suite of MARL benchmarks (22 scenarios, up to 30 agents). All experiments use at least 5 seeds (details in App. G.2.1).

### 5.1 Experimental Setup

**Environments.** HyperMARL is evaluated across 22 scenarios from five diverse MARL environments (Dispersion [5], Navigation [5], MAMuJoCo [37], SMAX [41], BPS [11]) (Table 2). These were specifically chosen to rigorously test performance across varying complexities, agent counts (2 to 30), and distinct behaviours (heterogeneous, homogeneous, or mixed). Full details in Appendix G.1.

Table 2: MARL environments for evaluating *HyperMARL*.

| Env. | Agents | Action | Behaviour |
|---|---|---|---|
| Dispersion | 4 | Discrete | Hetero. |
| Navigation | 2,4,8 | Continuous | Homo., Hetero., Mixed |
| MAMuJoCo | 2–17 | Continuous | Hetero. |
| SMAX | 2–20 | Discrete | Homo. |
| BPS | 15–30 | Discrete | Hetero. |

**Baselines.** We evaluate HyperMARL against modern parameter sharing (PS) and diversity-promoting baselines. Core PS comparisons use **FuPS+ID** and **NoPS**. For specialisation, we include *privileged* baselines: **DiCo** [6] (shared and non-shared weights, preset diversity levels); **HAPPO** [54] (shared critic, sequential actor updates); **Kaleidoscope** [28] (learnable masks, critic ensembles, diversity loss); and **SePS** (pre-training phase, agent clustering).

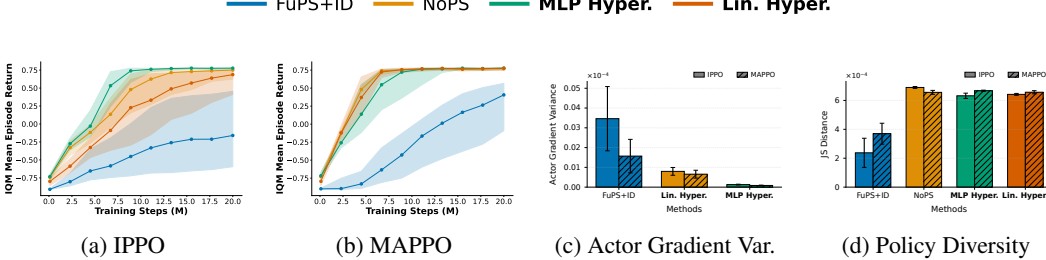

|     |     |     |     |
| :-: | :-: | :-: | :-: |
| (a) IPPO | (b) MAPPO | (c) Actor Gradient Var. | (d) Policy Diversity |

Figure 4: *Performance and gradient analysis.* **(a,b)** IPPO and MAPPO on Dispersion (20M timesteps) - IQM of Mean Episode Return with 95% bootstrap CIs: Hypernetworks match NoPS performance while FuPS struggle with specialisation. Interval estimates in App. H.3.1. **(c)** Actor gradient variance: Hypernetworks achieve lower gradient variance than FuPS+ID. **(d)** Policy diversity (SND with Jensen–Shannon distance): Hypernetworks achieve NoPS-level diversity while sharing parameters.

We use IPPO/MAPPO [13, 53] as the underlying algorithm for all methods except Kaleidoscope and SePS (see App. H.1 and App. H.2 for these results).

Adhering to best evaluation practices [36], we *use original codebases/hyperparameters and environments for which baselines were tuned*. HyperMARL uses identical observations and generates architectures of equivalent capacity to baselines. Training and evaluation (App. G.2.1) and hyperparameters (App. I) follow each baseline's original setup. We detail our baseline and environment selection criteria in Table 6, with architecture details in App. G.2.

**Measuring Policy Diversity.** To measure the diversity of the policies we System Neural Diversity (SND) [7] (Equation 5) with Jensen-Shannon distance (details in App. G.2.2).

## 5.2 Q1: Specialised Policy Learning

**Learning Diverse Behaviour (Dispersion)** Figures 4a and 4b show that FuPS variants (IPPO-FuPS, MAPPO-FuPS – (●)) can struggle to learn the diverse policies required by Dispersion (even when running for longer - Fig. 19), while their NoPS counterparts (IPPO-NoPS, MAPPO-NoPS–(●)) converge to the optimal policy, corroborating standard FuPS limitations to learn diverse behaviour. In contrast, HyperMARL (both linear and MLP variants) (●, ●) match NoPS performance, suggesting that a shared hypernetwork can effectively enable agent specialisation. *SND policy diversity measurements* (Fig. 4d) confirm FuPS variants achieve lower behavioural diversity than NoPS, while HyperMARL notably matches NoPS-level diversity.

**Gradient Variance.** HyperMARL (IPPO and MAPPO variants) also exhibits lower mean policy gradient variance than FuPS+ID across actor parameters (Fig. 4c). This aligns with their ability to learn diverse behaviours and supports the hypothesis that its gradient decoupling mechanism (Sec. 4.3) enhances training stability.

**Diversity at Complexity and Scale (MAMuJoCo).** In the challenging MAMuJoCo heterogeneous control tasks (Table 3), HyperMARL (MLP variant) is broadly competitive. Notably, unlike HAPPO and MAPPO (independent actors), HyperMARL uses a shared actor and parallel updates, and yet manages strong performance, even in the 17-agent Humanoid-v2 notoriously difficult heterogeneous task[54] (Fig. 5), matching methods that employ independent actors and sequential updates.

**Adaptability (Navigation).** Navigation tasks [5] assess adaptability to homogeneous, heterogeneous, and *mixed* goals (some agents have the same goals, others different). We compare HyperMARL with baselines including DiCo [6]. While using DiCo's optimal preset diversity for n=2 agents, we note that identifying appropriate diversity levels

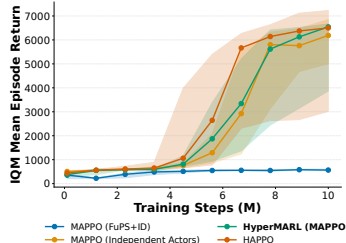

Figure 5: *17-agent Humanoid learning dynamics (IQM, 95% CI).* Hyper-MARL, utilising a shared actor architecture, outperforms MAPPO-FuPS (non-overlapping CIs) and matches the performance of methods employing non-shared or sequential actors. This challenging environment is recognised for its high variance in outcomes across different methods [54].

Table 3: *Mean episode return in MAMuJoCo for MAPPO variants(IQM, 95% CI).* HyperMARL achieves the highest IQM in 3/4 scenarios (bold), and is the only method with shared actors to demonstrate stable learning in the notoriously difficult 17-agent Humanoid environment (see Figure 5 for learning dynamics). * indicates CI overlap with the top score.

| Scenario | HAPPO | FuPS+ID | Ind. Actors | HyperMARL (Ours) |
|---|---|---|---|---|
| Humanoid-v2 17x1 | 6501.15* (3015.88, 7229.79) | 566.12 (536.36, 603.01) | 6188.46* (5006.13, 6851.74) | **6544.10** (3868.00, 6664.89) |
| Walker2d-v2 2x3 | 4748.06* (4366.94, 6230.81) | 4574.39* (4254.21, 5068.32) | 4747.05* (3974.76, 6249.58) | **5064.86** (4635.10, 5423.42) |
| HalfCheetah-v2 2x3 | 6752.40* (6130.42, 7172.98) | 6771.21* (6424.94, 7228.65) | 6650.31* (5714.68, 7229.61) | **7063.72** (6696.30, 7325.36) |
| Ant-v2 4x2 | 6031.92* (5924.32, 6149.22) | **6148.58** (5988.63, 6223.88) | 6046.23* (5924.62, 6216.57) | 5940.16* (5485.77, 6280.59) |

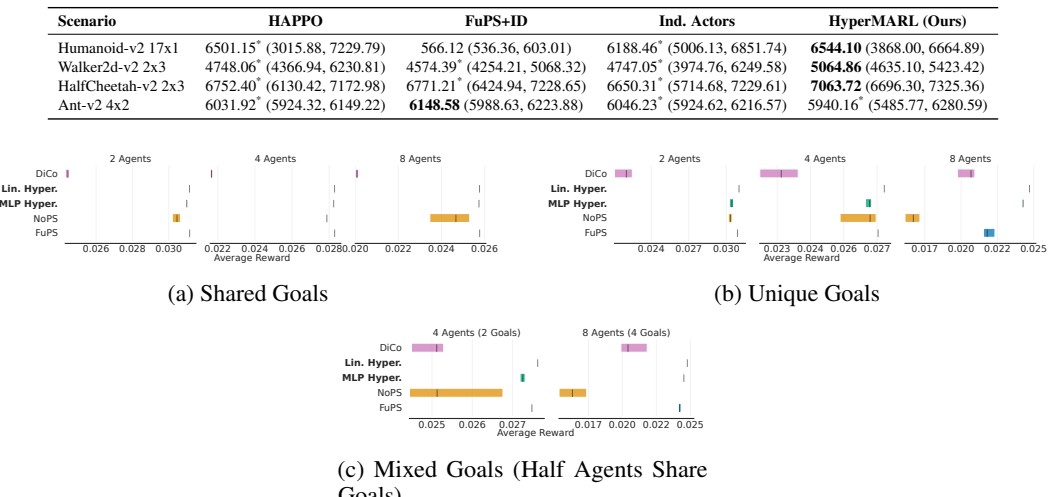

(a) Shared Goals      (b) Unique Goals

(c) Mixed Goals (Half Agents Share Goals)

Figure 6: *Average Reward (IQM, 95% CI) in Navigation for IPPO Variants.* HyperMARL adapts robustly across goal configurations—(a) shared, (b) unique, and (c) mixed. Both linear and MLP versions consistently match or outperform IPPO baselines and DiCo, with the margin widening as the number of agents grows. Sample-efficiency curves appear in App. H.5.

for DiCo with larger teams ($n > 2$) via hyperparameter sweeps proved challenging (see Tables 15 and 16).

Across all tested goal configurations (shared, unique, and mixed), HyperMARL consistently achieves strong performance (Figure 6). It generally matches or outperforms NoPS and FuPS, and outperforming DiCo. Interestingly, unlike in sparse-reward tasks like Dispersion, FuPS remains competitive with NoPS and HyperMARL in Navigation scenarios requiring diverse behaviours for smaller teams ($n \in \{2, 4\}$), likely due to Navigation's dense rewards. However, HyperMARL distinguishes itself as the strongest method for n=8 agents, highlighting its effectiveness in handling more complex coordination challenges.

### 5.3 Q2: Effectiveness in Homogeneous Tasks

**SMAX.** Finally, we evaluate HyperMARL (MLP) on SMAX, where recurrent FuPS is the established baseline [41, 53, 17]. Figure 7 shows while some FuPS variants might exhibit marginally faster initial convergence on simpler maps, HyperMARL achieves comparable final performance on all maps, using the same GRU backbone for partial observability. These results highlight two points: (i)

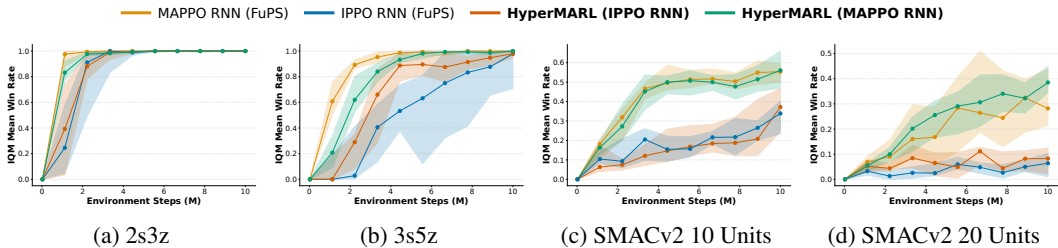

(a) 2s3z      (b) 3s5z      (c) SMACv2 10 Units      (d) SMACv2 20 Units

Figure 7: *IQM and 95% CI of mean win rate in SMAX.* Performance of FuPS Recurrent IPPO and MAPPO and HyperMARL (MLP) on SMAX. HyperMARL performs comparably to FuPS baselines across all environments, demonstrating its effectiveness in tasks requiring homogeneous behaviours and using recurrency. Interval estimates in Fig. 22 in App. H.6.

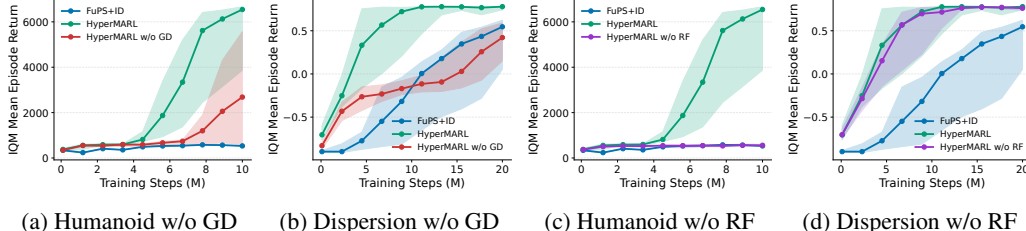

| (a) Humanoid w/o GD | (b) Dispersion w/o GD | (c) Humanoid w/o RF | (d) Dispersion w/o RF |

Figure 8: *Ablation results comparing HyperMARL to variants without gradient decoupling (w/o GD) and without reset fan in/out initialisation (w/o RF) across environments.* Gradient decoupling (a,b) is consistently critical across both environments, while initialisation scaling (c,d) shows greater importance in the complex Humanoid task but less impact in the simpler Dispersion environment.

HyperMARL is fully compatible with recurrent architectures essential under partial observability, and (ii) it has no intrinsic bias toward specialisation and can converge to homogeneous behaviour when it is optimal (also shown with strong same-goal Navigation performance (Fig. 6a)), even with large observation spaces and many agents.

**Summary.** Our empirical results confirm HyperMARL effectively addresses both research questions. For **Q1 (Specialisation)**, across Dispersion, MAMuJoCo, and Navigation, HyperMARL learned specialised policies, matched NoPS-level diversity and performance where FuPS+ID struggled, and scaled to complex, high-agent-count heterogeneous tasks. For **Q2 (Homogeneity)**, HyperMARL demonstrated competitive performance against strong FuPS baselines in SMAX and shared-goal Navigation, confirming its versatility.

## 6  Ablations and Embedding Analysis

### 6.1  Ablations: Gradient Decoupling and Initialisation Scaling

We ablate two critical components of HyperMARL: *gradient decoupling* (Sec. 4.3) and initialisation scaling (Sec. 4.2). In *HyperMARL w/o GD*, the hypernetwork is conditioned on $[o_t, e_t^i]$, coupling observation and agent-ID gradients. In *HyperMARL w/o RF*, we remove the reset fan-in/out scaling that aligns the scale of generated parameters $(\theta^i, \phi^i)$ with standard initialisers.

**Gradient decoupling is essential; initialisation scaling grows with complexity.** Figure 8 shows that removing GD consistently degrades performance across both *Humanoid-v2* (17 agents) and *Dispersion*, showing that GD is an essential component of HyperMARL. Removing RF reveals a task-dependent effect: it is critical on *Humanoid-v2*, consistent with hypernetwork initialisation results [8], but has a minor impact on *Dispersion*, indicating that principled initialisation becomes more vital with increased complexity. We provide additional ablations in App. H.7.

### 6.2  Analysis of Learned Agent Embeddings

Recall from 4.2 that each agent $i$ possesses an embedding $e^i$, which serves as input to the hypernetworks. For the MLP hypernetworks, these embeddings are learned end-to-end and are *orthogonally initialised*. Consequently, at initialisation (step 0), the pairwise cosine distance between any two distinct agent embeddings is $1.0$, mirroring the separability of one-hot IDs.

We probe how these embeddings $e^i$ adapt in the 4-agent Navigation task (same task as in Figure H.5) under two objectives with identical dynamics: (i) *same goal* (all agents navigate to a single shared target) and (ii) *different goals* (each agent navigates to its own unique target). At the end of training, we compute the mean pairwise cosine distance between agents' embeddings (lower values imply greater similarity) and compare it to the orthogonal baseline of $1.0$. (Cosine distance $= 1 -$ cosine similarity; $0 =$ identical direction, $1 =$ orthogonal,

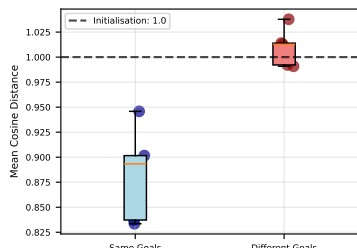

Figure 9: **Embedding similarity reflects task demands.** Mean pairwise cosine distance (dashed line $= 1.0$). Same goal: contraction ($0.882 \pm 0.042$). Different goals: near-orthogonal ($1.010 \pm 0.017$). Embeddings contract when a shared policy is optimal and maintain separation for specialisation.

2 = opposite). Figure 9 shows the per-seed distributions, with the 1.0 initialisation value as a dashed line.

**Agent embeddings adapt to task demands.** When behaviour should be homogeneous (*same goal*), embedding directions become significantly more aligned, the mean pairwise cosine distance *contracts* to $0.882 \pm 0.042$ (one-sample $t$-test vs 1.0: $p = 0.0079$). Conversely, when behaviour must differ (*different goals*), the embeddings *maintain their separation*, remaining near their orthogonal initialisation at $1.010 \pm 0.017$. These results demonstrate that the hypernetwork actively modulates the agent embeddings based on the task, promoting alignment for homogeneity while preserving separability for specialisation. We also conduct a sensitivity analysis on HyperMARL's hyperparameters in App F.5 and see that agent embedding size can be an important hyperparameter, one that could correspond to the task's diversity requirements.

# 7 Related Work

**Hypernetworks in RL and MARL.** Hypernetworks are effective in single-agent RL for meta-learning, multi-task learning, and continual learning [3, 4, 42, 21]. In MARL, QMIX [40] used hypernetworks (conditioned on a global state) to mix per-agent Q-values; however, each agent's own network remained a standard GRU. Parallel work, CASH [16], conditions a hypernetwork on local observations and predefined capability descriptors to target zero-shot generalisation across heterogeneous action spaces. By contrast, we use agent-conditioned hypernetworks in homogeneous action spaces, conditioning only on agent IDs/learned embeddings, and we explicitly decouple agent-conditioned from observation-conditioned gradients – a mechanism absent in CASH – which we find critical for specialisation.

**Variants of Parameter Sharing.** While Full Parameter Sharing (FuPS) is the most common approach, several other variants exist. Selective Parameter Sharing (SePS) [11] shares weights between similar groups of agents, identified by clustering agent trajectories during a pre-training phase. Pruning methods (SNP-PS, Kaleidoscope) [24, 28] split a single network into agent-specific subnetworks using learned agent masks. AdaPS [27] combines clustering and pruning masks to create shared policies for different groups of agents. Concurrent to our work, GradPS [38] identifies neurons with conflicting gradient updates, clones these neurons, and assigns each clone to a group of agents with low gradient conflict. Unlike these works, HyperMARL does not rely on pre-training (SePS), clustering algorithms requiring a preset number of clusters (SePS, GradPS, AdaPS), intricate pruning hyperparameters (SNP-PS, Kaleidoscope), auxiliary diversity losses (Kaleidoscope), or gradient conflict thresholds (GradPS).

**Learning Diverse Policies.** Shared parameters often limit policy diversity [11, 24, 17, 26]. Proposed solutions include: maximising mutual information between agent IDs and trajectories [26], role-based methods that assign distinct roles to agents [48, 49], best-response [39] methods and approaches that use structural modifications or constraints to induce diversity [24, 6, 28, 27]. Outside FuPS/NoPS, HAPPO [54] uses a non-shared centralised critic with individual actors updated sequentially to learn heterogeneous behaviours. In contrast to these works, HyperMARL does not alter the learning objective, use sequential updates or require preset diversity levels.

# 8 Conclusion

We investigated why standard parameter sharing fails at behavioural diversity, identifying that cross-agent gradient interference is critically exacerbated by coupling agent IDs with observations. We hypothesised that explicitly *decoupling* these gradients would enable adaptivity without prior complexities, and confirmed this using our *HyperMARL* approach. Our results show this decoupling enables adaptive behaviours (up to 30 agents) and is linked to reduced policy gradient variance. These findings establish gradient decoupling via HyperMARL as a versatile, principled approach for adaptive MARL. We discuss limitations in App. A, most notably parameter count, which can be remedied by parameter-efficient hypernetworks (e.g., chunked variants [47, 9]).

# 9   Acknowledgements

We would like to thank Samuel Garcin, Max Tamborski, Dave Abel, Timothy Hospedales, Trevor Mcinroe, Elliot Fosong, and Aris Filos-Ratsikas for fruitful discussions on early versions of this work. We also wish to acknowledge the anonymous reviewers for their constructive comments and feedback that helped strengthen this work.

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

## A   Limitations

Hypernetworks generate weights for target networks, which can lead to high-dimensional outputs and many parameters for deep target networks, particularly when using MLP-based hypernetworks. While HyperMARL uses more parameters than NoPS and FuPS for few agents, it scales almost constantly with agent count, an attractive property for large-scale MARL. Parameter efficiency could be improved through chunking techniques [47, 9], or low-rank weight approximations. This parameter overhead is often acceptable in RL/MARL given typically smaller actor-critic networks, and HyperMARL's favorable agent scaling (see App. F.3).

## B   Broader Impact

This paper presents work whose goal is to advance the field of Multi-Agent Reinforcement Learning. There are many potential societal consequences of our work, none which we feel must be specifically highlighted here.

## C   Specialised Policies and Environments

Specialisation plays a key role in MARL, yet remains under-defined, so we define *specialised environments* and *specialised policies*.

**Definition 1.** *An environment is specialised if the following both hold:*

1. ***Distinct Agent Policies.*** *The optimal joint policy $\boldsymbol{\pi}^*$ consists of at least two distinct agent policies, i.e., $\exists i, j \in \mathbb{I}$ such that $\pi^i \neq \pi^j$.*

2. ***Non-Interchangeability.*** *Any permutation $\sigma$ of the policies in $\boldsymbol{\pi}^*$, denoted as $\boldsymbol{\pi}^\sigma$, results in a weakly lower expected return:*

$$\mathbb{E}_{\mathbf{h} \sim \boldsymbol{\pi}^\sigma}[G(\mathbf{h})] \leq \mathbb{E}_{\mathbf{h} \sim \boldsymbol{\pi}^*}[G(\mathbf{h})],$$

   *with strict inequality if the joint policies are* non-symmetric *(i.e., swapping any individual policy degrades performance).*

For example, consider a *specialised environment* such as a football game, optimal team performance typically requires players in distinct roles (e.g., "attackers," "defenders"). Permuting these roles (i.e., exchanging their policies) would typically lead to suboptimal results. Here, agents develop *specialised policies* by learning distinct, complementary behaviours essential for an optimal joint policy. While agents with heterogeneous capabilities (e.g., different action spaces) are inherently specialised, homogeneous agents can also learn distinct policies. Such environments are analysed in Sections E.1 and 5.2.

## D   Measuring Behavioural Diversity

### D.1   Quantifying Team Diversity

We quantify policy diversity using System Neural Diversity (SND) [7], which measures behavioural diversity based on differences in policy outputs:

$$\text{SND}\left(\left\{\pi^i\right\}_{i \in \mathbb{I}}\right) = \frac{2}{n(n-1)|\mathcal{O}|} \sum_{i=1}^{n} \sum_{j=i+1}^{n} \sum_{o \in \mathcal{O}} D\left(\pi^i(o), \pi^j(o)\right). \tag{5}$$

where $n$ is the number of agents, $\mathcal{O}$ is a set of observations typically collected via policy rollouts, $\pi^i(o_t)$ and $\pi^j(o_t)$ are the outputs of policies $i$ and $j$ for observation $o_t$, and $D$ is our distance function between two probability distributions.

In contrast to [7], we use Jensen-Shannon Distance (JSD) [14, 29] as $D$, rather than the Wasserstein metric [46]. As shown in Appendix D.2, JSD is a robust metric for both continuous and discrete cases, and provides a more reliable measure of policy distance.

## D.2   Finding a Suitable Distance Function for Policy Diversity

The choice of distance function $D$ in Equation 5 is crucial for accurately measuring policy diversity. In MARL, policies are often represented as probability distributions over actions, making the choice of distance function non-trivial.

[6] use the Wasserstein metric for continuous policies [46] as distance function $D$, while [31] use the total variation distance for discrete policies. For discrete policies, Wasserstein distance would require a cost function representing the cost of changing from one action to another, which might not be trivial to come up with. On the other hand, although well-suited for discrete policies, TVD might miss changes in action probabilities because it measures the largest difference assigned to an event (i.e. action) between two probability distributions.

We consider a simple example to illustrate this point. Suppose we have two policies $\pi^1$ and $\pi^2$ with action probabilities as shown in Figure 10. $\pi^1$ stays constant, while $\pi^2$ changes gradually over timesteps. We see that even as $\pi^2$ changes over time, the $TVD(\pi^1, \pi^2)$ between $\pi^1$ and $\pi^2$ remains constant. This is because TVD only measures the largest difference between the two distributions, and does not consider the overall difference between them. On the other hand, the Jensen-Shannon distance (JSD) [14], which is the square root of the Jensen-Shannon divergence, does not have this problem as it is a smooth distance function. Furthermore, it satisfies the conditions for being a metric – it is non-negative, symmetric, and it satisfies the triangle inequality.

For continuous policies, as shown in Figure 11, JSD exhibits similar trends to the Wasserstein distance. Since JSD is a reasonable metric for both continuous and discrete probability distributions, we will use it as the distance metric for all experiments and propose it as a suitable distance function for measuring policy diversity in MARL.

We also summarise the properties of the various distance metrics in Table 4.

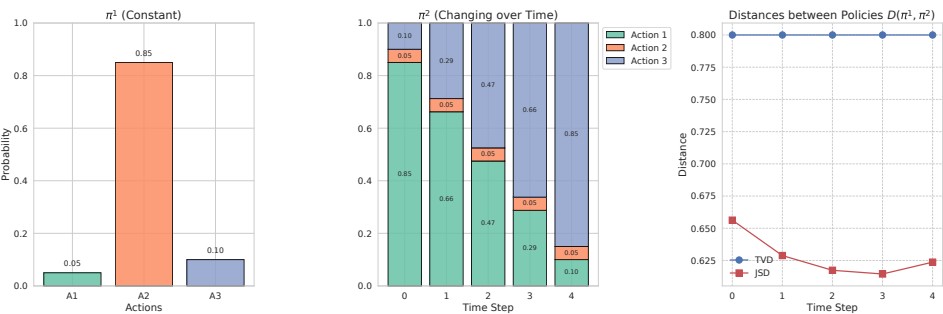

Figure 10: Gradual changes in $\pi^2$, result in gradual changes in the Jensen-Shannon distance (JSD), while the Total Variation Distance (TVD) can miss changes in action probabilities.

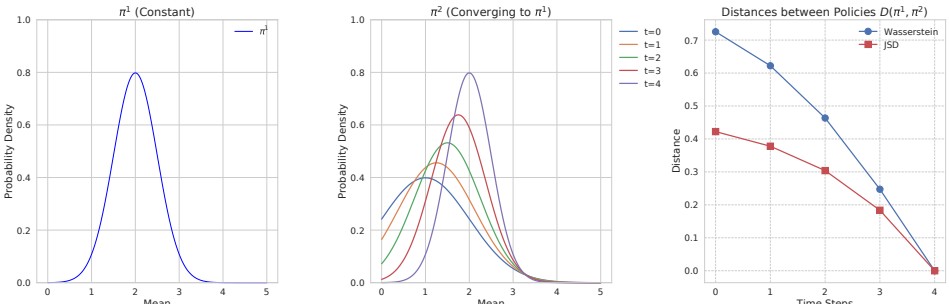

Figure 11: Jensen-Shannon distance (JSD) trends similarly to Wasserstein distance when we have continuous policies.

| Method | Kinds of Actions | Metric | Smooth | Formula |
|---|---|---|---|---|
| Wasserstein Distance [46] | Continuous* | Metric | Yes | $W(p,q) = \left( \inf_{\gamma \in \Gamma(p,q)} \int_{\mathbb{R} \times \mathbb{R}} \|x-y\| \, d\gamma(x,y) \right)^{1/p}$ |
| Total Variation Distance | Discrete | Metric | No | $TV(p,q) = \frac{1}{2} \sum_x \|p(x) - q(x)\|$ |
| Jensen-Shannon Divergence [29] | Both | Divergence | Yes | $JSD(p \parallel q) = \frac{1}{2} D_{KL}(p \parallel m) + \frac{1}{2} D_{KL}(q \parallel m), \ m = \frac{1}{2}(p+q)$ |
| Jensen-Shannon Distance [14] | Both | Metric | Yes | $\sqrt{JSD(p \parallel q)}$ |

Table 4: Measures for Policy Diversity

# E    Specialisation and Synchronisation Games

To study the challenges of specialisation and coordination in an isolated setting, we introduce the Specialisation and Synchronisation Games, drawing inspiration from a version of the XOR game [17], VMAS's Dispersion [5] and coordination and anti-coordination games in game theory [35]. These environments encourage agents to take distinct actions (Specialisation) or take identical actions (Synchronisation). Despite their deceptively simple payoff structure, these games present substantial learning challenges – non-stationary reward distributions driven by others' adapting behaviours and in their temporal extension, the joint observation spaces grows exponentially with the number of agents.

## E.1    Specialisation and Synchronisation Games Description

**Specialisation Game.** Agents are encouraged to choose *distinct* actions. In the simplest setting, it is a two-player matrix game where each agent selects between two actions ($A$ or $B$). As shown in Figure 2a, agents receive a payoff of $1.0$ when their actions differ (creating two pure Nash equilibria on the anti-diagonal) and $0.5$ when they match. This structure satisfies Definition 1, since optimal joint policies require complementary, non-identical strategies. There is also a symmetric mixed-strategy equilibrium in which each agent plays $A$ and $B$ with probability $0.5$.

**Synchronisation Game.** Conversely, agents are encouraged to coordinate and choose *identical* actions. The payoff matrix inverts the Specialisation game's structure, agents receive 1 for matching actions and $0.5$ for differing ones. This creates two pure Nash equilibria along the diagonal of the payoff matrix (Figure 2b), and incentivises uniform behaviour across agents.

$N$**-Agent Extension.** Both games naturally scale to $n$ agents and $n$ possible actions. In Specialisation, unique actions receive a payoff of $1.0$, while selecting the same action receives payoffs of $\frac{1}{k}$, where $k$ is the number of agents choosing that action. In contrast, in Synchronisation, agents receive maximum payoffs ($1.0$) only when all actions match. For partial coordination, rewards follow a hyperbolic scale, $\frac{1}{n-k+1}$, encouraging agents to align their choices. Visualisations in Figure 2 and detailed reward profiles appear in Figure 12.

## E.2    General-$n$ Payoff Definitions

Both games generalise naturally to $n$ agents and $n$ possible actions. We show the reward profiles for $n = 5$ agents in Figure 12.

Let $\mathbf{a} = (a^1, \ldots, a^n) \in \{1, \ldots, n\}^n$ and $k_a = \left| \{ j : a^j = a \} \right|$ be the joint action profile and the count of agents choosing action $a$, respectively.

**Temporal Extension.** To model sequential decision-making, we extend each normal-form game into a repeated Markov game, where the state at time $t$ is the joint action at time $t - 1$. At each step $t$ all agents observe $a^{t-1}$, choose $a_i^t$, and receive the original Specialisation or Synchronisation payoff. This repeated setup isolates how agents adapt based on past joint actions, exposing temporal patterns of specialisation and coordination.

### E.2.1    Specialisation Game

The reward is formulated as follows:

$$r_{\text{spec}}^i(\mathbf{a}) = \begin{cases} 1.0, & \text{if } k_{a^i} = 1 \quad \left(\text{unique action}\right); \\ \dfrac{1}{k_{a^i}}, & \text{if } k_{a^i} > 1 \quad \left(\text{shared action}\right). \end{cases}$$

### E.2.2    Synchronisation Game

The reward is formulated as follows:

$$r_{\text{sync}}^i(\mathbf{a}) = \frac{1}{n - k_{a^i} + 1},$$

so that $r_{\text{sync}}^i = 1.0$ when $k_{a^i} = n$ (all agents select the same action), and otherwise follows a hyperbolic scale encouraging consensus.

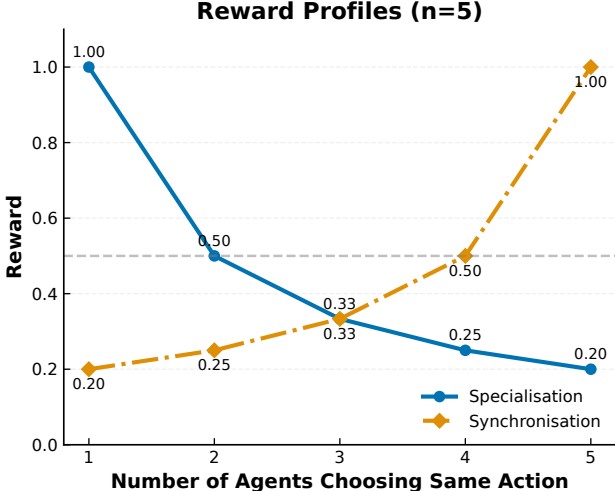

Figure 12: Reward profiles for the Specialisation (blue) and Synchronisation (orange) games with $n = 5$ agents. In the Specialisation game, an agent's payoff peaks when it selects a unique action, and then decays as when actions are shared. In the Synchronisation game, payoffs follow a hyperbolic scale $1/(n - k + 1)$, reaching maximum only under full consensus, thereby incentivising alignment.

### E.3 Proof that FuPS cannot represent the optimal policy in the two-player Specialisation Game

**Theorem 1.** *A stochastic, shared policy without agent IDs cannot learn the optimal behaviour for the two-player Specialisation Game.*

*Proof.* Let $\pi$ be a shared policy for both agents, and let $\alpha = \mathbb{P}(a_i = 0)$ represent the probability of any agent choosing action 0. The expected return of $\pi$ for each agent is:

$$E[R(\pi)] = \mathbb{P}(a_0 = 0, a_1 = 0) \cdot 0.5 + \mathbb{P}(a_0 = 0, a_1 = 1) \cdot 1 \tag{6}$$

$$+ \mathbb{P}(a_0 = 1, a_1 = 0) \cdot 1 + \mathbb{P}(a_0 = 1, a_1 = 1) \cdot 0.5 \tag{7}$$

$$= 0.5\alpha^2 + 2\alpha(1 - \alpha) + 0.5(1 - \alpha)^2 \tag{8}$$

$$= -\alpha^2 + \alpha + 0.5 \tag{9}$$

$$= -(\alpha - 0.5)^2 + 0.75 \tag{10}$$

Thus, $E[R(\pi)] \leq 0.75 < 1$ for all $\alpha \in [0, 1]$, with the maximum at $\alpha = 0.5$. Therefore, a shared policy cannot achieve the optimal return of 1, confirming the need for specialised behaviour to optimise rewards. $\square$

### E.4 Measuring Agent Gradient Conflict

We measure *gradient conflicts*, via the cosine similarity between agents' gradients $\cos\big(g_t^{(i)}, g_t^{(j)}\big) = \frac{\langle g_t^{(i)}, g_t^{(j)} \rangle}{\|g_t^{(i)}\|\|g_t^{(j)}\|}$, where $g_t^{(i)} = \nabla_\theta \mathcal{L}^{(i)}(\theta_t)$.

### E.5 Empirical Results in N-player Specialisation and Synchronisation Normal-Form Game

To assess this limitations of FuPS and NoPS in practice, we compare three REINFORCE [51] variants in both games with $n = 2, 4, 8, 16, 32$ agents: NoPS (No Parameter Sharing), FuPS (Fully Parameter Sharing), and FuPS+ID (FuPS with one-hot agent IDs). All policies use single-layer neural networks with controlled parameter counts (see Appendix I for details).

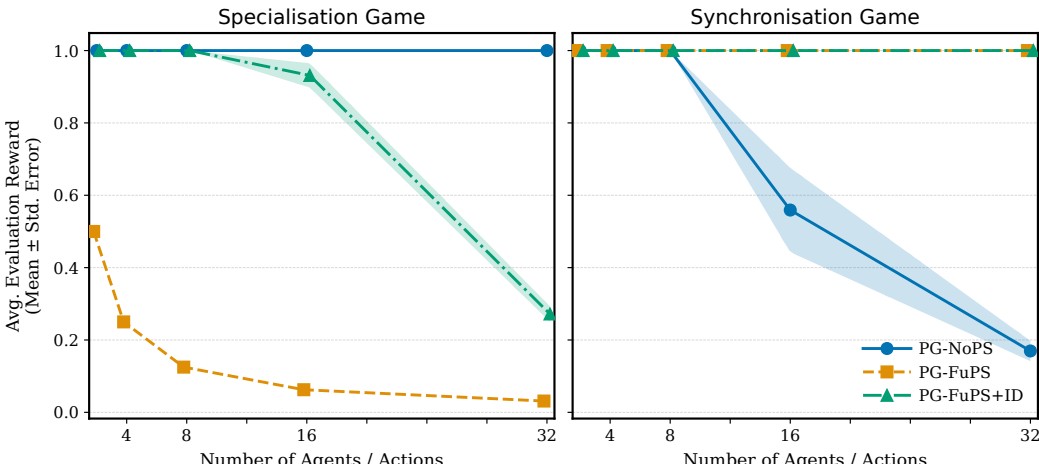

Figure 13: Mean evaluation reward (**mean ± standard error**) as a function of the number of agents/actions in the **Specialisation** (left) and **Synchronisation** (right) games. In the Specialisation game, vanilla policy gradients (PG, i.e. REINFORCE) with FuPS collapse as the team grows, whereas our identity-aware variant (PG-FuPS+ID) retains near-optimal performance. In the Synchronisation game, PG-NoPS performs well at small scales but degrades with more agents, while both PG-FuPS and PG-FuPS+ID remain at optimal reward across all scales.

---

**Algorithm 1** HyperMARL

---

1: **Input:** Number of agents $n$, number of training iterations $K$, MARL algorithm parameters (e.g., MAPPO-specific hyperparameters)
2: **Initialise:**
3:     Hypernetwork parameters $\psi, \varphi$ {Ensure $\theta^i$ and $\phi^i$ follow standard initialization schemes, e.g., orthogonal}
4:     Agent embeddings $\{e^i\}_{i=1}^n$ {One-hot or orthogonally initialised learnable parameters}
5: **Output:** Optimized joint policy $\boldsymbol{\pi}$
6: **for** each training iteration $k = 0, 1, \ldots, K - 1$ **do**
7:     **for** each agent $i = 1, \ldots, n$ **do**
8:         $\theta^i \leftarrow h_\psi^\pi(e^i)$ {Policy parameters}
9:         $\phi^i \leftarrow h_\varphi^V(e^i)$ {Critic parameters}
10:     **end for**
11:     Interact with environment using $\{\pi_{\theta^i}\}_{i=1}^n$ to collect trajectories $\mathcal{D}$
12:     Compute shared loss $\mathcal{L}$ from $\mathcal{D}$, using $\{V_{\phi^i}\}_{i=1}^n$ {Standard RL loss function}
13:     Update $\psi, \varphi$, and $e$ by minimizing $\mathcal{L}$ {Optimise parameters.}
14: **end for**
15: **Return** $\boldsymbol{\pi} = (\pi^1, \ldots, \pi^n)$

---

# F HyperMARL Details

## F.1 HyperMARL Pseudocode

In Algorithm 1, we present the pseudocode for HyperMARL, with HyperMARL-specific steps highlighted in blue. HyperMARL leverages hypernetworks to dynamically generate the parameters of both actor and critic networks. The weights of the hypernetworks and the agent embeddings are automatically updated through automatic differentiation (autograd) based on the computed loss. Additionally, Figure 1 provides a visual representation of the HyperMARL architecture.

## F.2 Variance of the HyperMARL Gradient Estimator

**Unbiased estimator.** Following from Equation 4, we can write the unbiased estimator for Hyper-MARLs gradient as follows:

$$
\widehat{g}_{\text{HM}} = \sum_{i=1}^I \underbrace{\nabla_\psi h_\psi^\pi(e^i)}_{\mathbf{J}_i} \cdot \underbrace{\left( \frac{1}{B} \sum_{b=1}^B \sum_{t=0}^{T-1} A(\mathbf{h}_t^{(b)}, \mathbf{a}_t^{(b)}) \nabla_{\theta^i} \log \pi_{\theta^i}(a_t^{i,(b)} \mid h_t^{i,(b)}) \right)}_{\widehat{Z}_i} = \sum_{i=1}^I \mathbf{J}_i \, \widehat{Z}_i,
$$

(HM')

where $B$ trajectories $\{\tau^{(b)}\}_{b=1}^B$ are sampled i.i.d. and $\widehat{Z}_i$ is the empirical analogue of the observation-conditioned factor.

**Assumptions.** (A1) trajectories are i.i.d.; (A2) all second moments are finite; (A3) $\psi, \theta, e^i$ are fixed during the backward pass.

**Variance expansion.** Since each $\mathbf{J}_i$ is deterministic under (A3), we may factor them outside the covariance:

$$
\begin{aligned}
\text{Var}(\widehat{g}_{\text{HM}}) &= \text{Cov}\Big( \sum_i \mathbf{J}_i \widehat{Z}_i, \ \sum_j \mathbf{J}_j \widehat{Z}_j \Big) && (\text{by def. } \text{Var}(X) = \text{Cov}(X, X)) \\
&= \sum_{i,j} \text{Cov}(\mathbf{J}_i \widehat{Z}_i, \ \mathbf{J}_j \widehat{Z}_j) && (\text{bilinearity of Cov}) \\
&= \sum_{i,j} \mathbf{J}_i \ \text{Cov}(\widehat{Z}_i, \widehat{Z}_j) \ \mathbf{J}_j^\top && (\text{pull deterministic matrices out of Cov})
\end{aligned}
$$

(11)

Equation (11) makes explicit that all trajectory-induced covariance is captured $\mathrm{Cov}(\widehat{Z}_i, \widehat{Z}_j)$, while the agent-conditioned Jacobians $\mathbf{J}_i$ remain trajectory noise-free.

**Mini-batch update and covariance.** Let $\widehat{Z}_i$ be the unbiased mini-batch estimate of $Z_i$ and $\widehat{g}_{\mathrm{HM}} = \sum_i \mathbf{J}_i \widehat{Z}_i$ the stochastic update. Because every $\mathbf{J}_i$ is deterministic (wrt. to mini-batch),

$$\mathrm{Var}(\widehat{g}_{\mathrm{HM}}) = \sum_{i,j} \mathbf{J}_i \ \mathrm{Cov}(\widehat{Z}_i, \widehat{Z}_j) \ \mathbf{J}_j^\top, \tag{12}$$

(derivation in Appendix F.2). Equation (12) shows that HyperMARL first averages noise *within each agent* ($\widehat{Z}_i$) and only then applies $\mathbf{J}_i$. FuPS+ID, by contrast, updates the shared weights $\theta$ with every raw sample $A \nabla_\theta \log \pi_\theta[h, \mathrm{id}]$, leaving observation noise and agent ID entangled and making it susceptible to gradient interference [11, 54].

### F.3   Scalability and Parameter Efficiency

Hypernetworks generate weights for the target network, which can lead to high-dimensional outputs and many parameters for deep target networks. This challenge is amplified in MLP-based hypernetworks, which include additional hidden layers. Figure 14 shows scaling trends:

- **NoPS** and **linear hypernetworks**: Parameter count grows linearly with the number of agents.

- **FuPS**: More efficient, as growth depends on one-hot vector size.

- **MLP hypernetworks**: Scale better with larger populations, since they only require embeddings of fixed size for each new agent.

To reduce parameter count, techniques like shared hypernetworks, chunked hypernetworks [47, 9], or producing low-rank weight approximations, can be used. While naive implementations are parameter-intensive, this might be less critical in RL and MARL which commonly have smaller actor-critic networks. Moreover, HyperMARL's near-constant scaling with agents suggests strong potential for large-scale MARL applications.

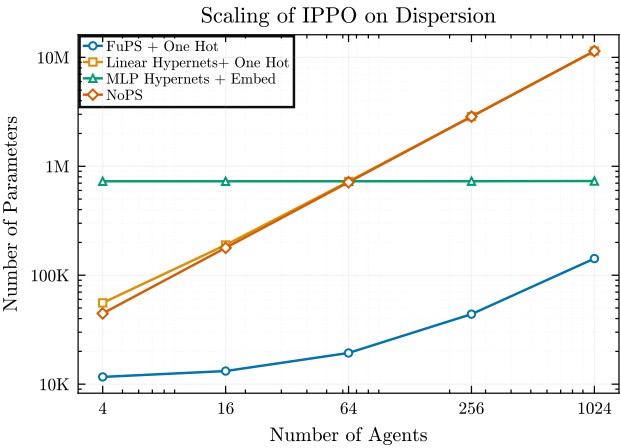

Figure 14: Parameter scaling for IPPO variants with increasing agents (4 to 1024). MLP Hypernetworks scale nearly constantly, while NoPS, Linear Hypernetworks, and FuPS+One-Hot grow linearly. Log scale on both axes.

To isolate the effects of parameter count, we scaled the FuPS networks (Figure 15) to match the number of trainable parameters in HyperMARL. Despite generating 10x smaller networks, HyperMARL consistently outperforms FuPS variants, showing its advantages extend beyond parameter count.

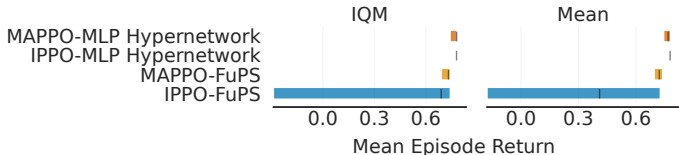

Figure 15: Dispersion performance with four agents. FuPS variants match HyperMARL in parameter count but still underperform.

## F.4 Speed and Memory Usage

We examine the computational efficiency of HyperMARL compared to NoPS and FuPS by measuring inference speed (Figure 16a) and GPU memory usage (Figure 16b) as we scale the number of agents. The benchmarks were conducted using JAX on a single NVIDIA GPU (T4) with a recurrent (GRU-based) policy architecture. All experiments used fixed network sizes (64-dimensional embeddings and hidden layers) with a batch size of 128 and 64 parallel environments, allowing us to isolate the effects of varying agent count. Each measurement represents the average of 100 forward passes per configuration, with operations repeated across 10 independent trials.

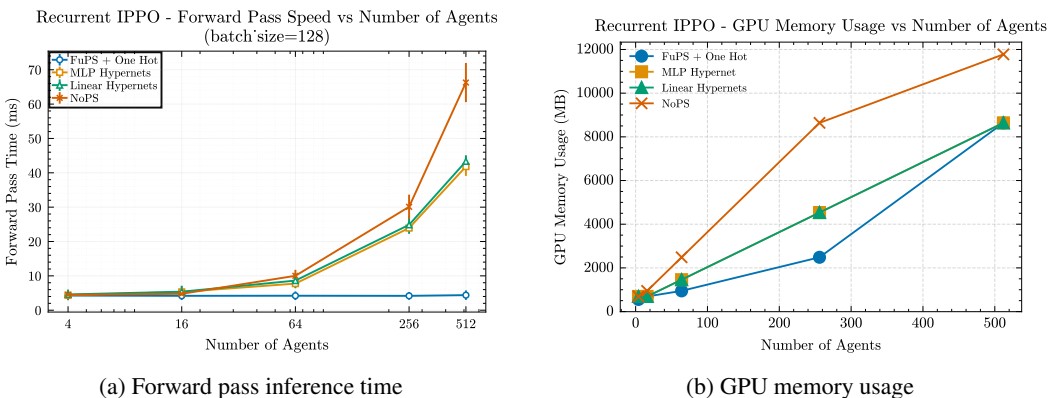

(a) Forward pass inference time               (b) GPU memory usage

Figure 16: Computational efficiency scaling with number of agents. HyperMARL offers a balance between NoPS and FuPS. Notably, in real-world deployments, NoPS incurs additional data transfer and synchronisation costs not reflected here, further widening the efficiency gap.

The results demonstrate that HyperMARL offers a balance between the extremes represented by NoPS and FuPS. In practice, NoPS incurs additional data transfer and update costs, widening the efficiency gap.

## F.5 Sensitivity Analysis

We run a sensitivity analysis on the HyperMARL's hyperparameters in the 20-agent SMAX scenario with recurrent MAPPO, a challenging setting for optimisation. The results are shown Table 5.

**Findings.**

- Similarly to FuPS, and in general, deep RL methods like PPO, the learning rate is an important parameter for performance.
- **Agent embedding size is an important hyperparameter.** In these results, a smaller embedding size outperformed a larger one. This could be due to the homogeneous nature of some SMAX tasks, where with smaller embeddings, it could be easier to learn similar agent embeddings and hence similar behaviours. This suggests the embedding size could correspond to the task's diversity requirements.
- Other parameters, such as the width, have limited impact beyond a modest size.

Table 5: IQM and 95% CI of mean win rate across 5 seeds for the 20-agent SMAX scenario with recurrent MAPPO. **Bold** indicates the highest IQM score; * indicates scores whose 95% confidence intervals overlap with the highest score.

| Method | Embedding Dim | Hidden Dims | LR 0.0001 | LR 0.0003 | LR 0.0005 |
|---|---|---|---|---|---|
| FuPS | N/A | N/A | 0.1067 (0.0513, 0.2128) | 0.3053 (0.2041, 0.4769) | *0.4213 (0.3623, 0.4800) |
| HyperMARL | 4 | 16 | 0.0799 (0.0250, 0.1190) | 0.2946 (0.1143, 0.3778) | **0.4455** (0.3443, 0.5692) |
| | | 64 | 0.1233 (0.0750, 0.2162) | 0.3338 (0.2692, 0.4286) | *0.3765 (0.3281, 0.4107) |
| | | 128 | 0.1147 (0.0270, 0.1842) | *0.3787 (0.3509, 0.4603) | 0.3327 (0.2931, 0.3548) |
| | | 64, 64 | 0.1781 (0.1628, 0.2075) | *0.3498 (0.2807, 0.4500) | 0.1944 (0.1268, 0.3077) |
| HyperMARL | 64 | 16 | 0.1109 (0.0256, 0.2093) | 0.3198 (0.2353, 0.3934) | *0.3483 (0.2581, 0.4308) |
| | | 64 | 0.1360 (0.0303, 0.2821) | 0.1191 (0.0517, 0.1964) | 0.1155 (0.0794, 0.1500) |
| | | 128 | 0.1193 (0.0750, 0.1579) | 0.1036 (0.0517, 0.1594) | 0.1106 (0.0455, 0.1587) |
| | | 64, 64 | 0.1353 (0.0541, 0.1818) | 0.1283 (0.0526, 0.2258) | 0.0860 (0.0345, 0.1343) |

# G Experiment Details

## G.1 Environments

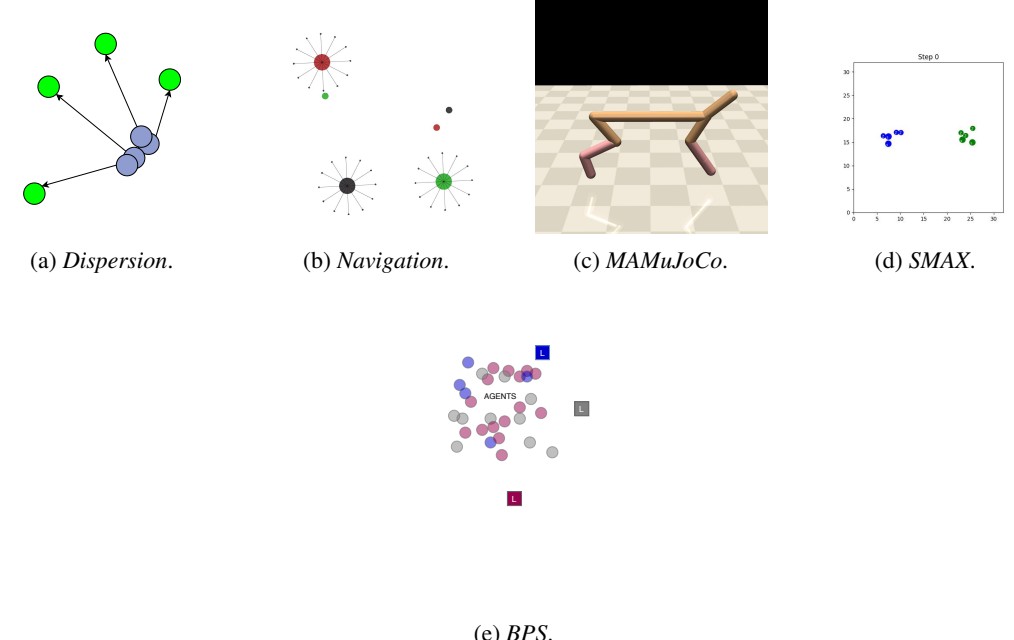

(a) *Dispersion.*  (b) *Navigation.*  (c) *MAMuJoCo.*  (d) *SMAX.*

(e) *BPS.*

Figure 17: Multi-Agent environments used in our experiments.

**Dispersion (VMAS) [5]**: A 2D environment where four agents collect unique food particles. This task requires specialised *heterogeneous* behaviours and resembles the Specialisation Game from Section E.1.

**Navigation (VMAS) [5]**: Agents navigate in a 2D space to assigned goals, receiving dense rewards based on proximity. Configurations include shared goals (*homogeneous*), unique goals (*heterogeneous*), and *mixed* goals, where some agents share goals while others have unique ones.

**Multi-Agent MuJoCo (MAMuJoCo) [37]**: A multi-agent extension of MuJoCo, where robot body parts (e.g., a cheetah's legs) are modelled as different agents. Agents coordinate to perform efficient motion, requiring *heterogeneous* policies [54].

**SMAX (JaxMARL) [41]**: Discrete action tasks with 2 to 20 agents on SMACv1- and SMACv2-style maps. FuPS baselines have been shown optimal for these settings [53, 17] indicating *homogeneous* behaviour is preferred here.

**Blind-Particle Spread (BPS) [11]** Blind-Particle Spread (BPS) is a modified variant of the Multi-Agent Particle Environment (MPE) Simple Spread task [30] with 15 to 30 agents. The environment contains landmarks of different colours, and each agent is assigned a colour. Agents are 'blind' as they cannot observe their own assigned colour or the colours of other agents. They must infer the correct landmark to navigate towards based only on their observations and rewards. The number of colours thus dictates the number of distinct roles or behaviours the team must learn to successfully complete the task.

## G.2 HyperMARL Architecture Details

For the Dispersion and Navigation results (Sec. 5.2) we use feedforward architectures, where we use HyperMARL to generate both the actor and critic networks. For the MAPPO experiments in Section 5.2, for fairness in comparisons with HAPPO and MAPPO, we maintain the centralised critic conditioned on the global state and only use HyperMARL to generate the weights of the actors. For the recurrent IPPO experiments in Section 5.3, HyperMARL only generates the actor and critic feedforward weights, not the GRU weights.

Table 6: Baseline Methods Selection and Justification. Selected methods (✓) were chosen based on their relevance to parameter sharing and specialisation/generalisation in MARL, while excluded methods (✗) did not align with our research objectives or had implementation constraints. **Our experimental design systematically addresses key questions on agent specialisation and homogeneity, therefore we selected baselines with demonstrated strong performance in their respective settings, ensuring fair and rigorous comparison.**

| Method | Category | Selected | Justification & Experimental Settings |
|---|---|---|---|
| **IPPO** [13] (NoPS, FuPS+ID) | NoPS/FuPS | ✓ | Established MARL baseline implementing both independent (NoPS) and fully shared (FuPS+ID) policy configurations. *Tested in:* Dispersion, Navigation, SMAX (two SMACv1 maps and two SMACv2 maps, with 10 and 20 agents). |
| **MAPPO** [53] (NoPS, FuPS+ID) | NoPS/FuPS | ✓ | Strong baseline with centralized critics for both NoPS and FuPS+ID architectures. *Tested in:* Dispersion, MAMuJoCo, SMAX (two SMACv1 maps and two SMACv2 maps, with 10 and 20 agents). |
| **DiCo** [6] | Architectural Diversity | ✓ | Provides comparison with a method employing preset diversity levels that balances shared and non-shared parameters. *Tested in:* Dispersion and Navigation (as per original paper). Original hyperparameters used for $n = 2$ agents; parameter sweeps conducted for $n > 2$ to identify optimal diversity levels. |
| **HAPPO** [54] | Sequential Updates | ✓ | Enables comparison with a method designed for heterogeneous behaviours using sequential policy updates with agent-specific parameters. *Tested in:* MAMuJoCo, selecting 4/6 scenarios from the original paper, including the challenging 17-agent humanoid task. Walker and Hopper variants were excluded as MAPPO and HAPPO performed similarly in these environments. |
| **Kaleidoscope** [28] | Architectural Pruning | ✓ | Implemented for off-policy evaluation using its MATD3 implementation with tuned MaMuJoCo hyperparameters. *Tested in:* MAMuJoCo environments Ant-v2, HalfCheetah-v2, Walker2d-v2 (overlapping with our IPPO experiments), and Swimmer-v2-10x2 (highest agent count variant). Included to evaluate HyperMARL's competitiveness against a method with ensemble critics and diversity loss, in an off-policy setting. |
| **SePS** [11] | Selective Parameter Sharing | ✓ | Although this requires a pre-training phase, it is a strong baseline for parameter sharing approaches. *Tested in:* Blind-Particle Spread environments, with 15 to 30 agents. |
| **SEAC** [12] | Shared Experience | ✗ | Focuses primarily on experience sharing rather than parameter sharing architecture, falling outside our research scope. |
| **CDAS** [26] | Intrinsic Reward | ✗ | Only implemented for off-policy methods and has been demonstrated to underperform FuPS/NoPS architectures [17], making it less suitable for our primary on-policy comparisons. |
| **ROMA/RODE** [48, 49] | Role-based | ✗ | Shows limited practical performance advantages in comparative studies [12], suggesting other baselines provide more rigorous comparison points. |
| **SNP-PS** [24] | Architectural Pruning | ✗ | No publicly available implementation, preventing direct, reproducible comparison. |

### G.2.1 Training and Evaluation

- **Training:**

    - For Dispersion (5.2), we run 10 seeds and train for 20 million timesteps.
    - For Navigation (5.2), SMAX (5.3), and MaMuJoCo (5.2), we run 5 seeds and train for 10 million timesteps, consistent with the baselines.

- For Blind-Particle Spread (BPS), we run 5 seeds and train for 20 million timesteps, consistent with baselines.

- **Evaluation:**

    - For Dispersion (5.2), evaluation is performed every 100k timesteps across 32 episodes.
    - For Navigation (5.2), following the baselines, evaluation is performed every 120k timesteps across 200 episodes.
    - For SMAX (5.3), evaluation is performed every 500k timesteps across 32 episodes.
    - For MaMuJoCo (5.2), following the baselines, evaluation is performed every 25 training episodes over 40 episodes.

### G.2.2 Measuring Policy Diversity Details

We measure team diversity using the System Neural Diversity (SND) metric (Equation 5 [7], details Section D) with Jensen-Shannon distance. SND ranges from 0 (identical policies across all agents) to 1 (maximum diversity). We collect a dataset of observations from IPPO-NoPS and IPPO-FuPS policies checkpointed at 5 and 20 million training steps. Each policy is rolled out for 10,000 episodes, generating 16 million observations. We then sample 1 million observations from this dataset to calculate the SND for each method tested.

Table 7: *Mean episode return in MAMuJoCo for off-policy MATD3 variants.* IQM of the mean episode returns with 95% stratified bootstrap CI. **Bold** indicates the highest IQM score; * indicates scores whose confidence intervals overlap with the highest. Although Kaleidoscope employs an ensemble of five critics and an explicit diversity loss, HyperMARL (using a standard MATD3 setup with two critics) achieves competitive results without these additional mechanisms.

| Environment | Ind. Actors, Shared Critic | Kaleidoscope | HyperMARL |
|---|---|---|---|
| Ant-v2 | 5270.38 (4329.73, 5719.78) | **6160.70** (5798.02, 6463.83) | *5886.58 (5840.00, 5920.66) |
| HalfCheetah-v2 | *6777.04 (3169.11, 8233.94) | *6901.00 (3609.73, 8192.38) | **7057.44** (3508.70, 8818.11) |
| Walker2d-v2 | *5771.87 (5144.84, 8103.34) | *6664.32 (5408.95, 8828.11) | **7057.68** (5976.50, 8166.09) |
| Swimmer-v2-10x2 | *453.74 (427.24, 487.86) | *462.48 (444.22, 475.64) | **465.91** (410.82, 475.77) |

# H    Detailed Results

## H.1    Comparison with Kaleidoscope using Off-Policy – MATD3

Our comparison with Kaleidoscope [28], mentioned in Section 5, is conducted using off-policy methods due to its original design. Kaleidoscope incorporates intricate mechanisms (e.g., learnable masks, an ensemble of five critics, a specific diversity loss) and numerous specialised hyperparameters (e.g., for critic ensembling: 'critic_deque_len', 'critic_div_coef', 'reset_interval'; for mask and threshold parameters: 'n_masks', 'threshold_init_scale', 'threshold_init_bias', 'weighted_masks', 'sparsity_layer_weights', etc.). Porting this full complexity to an on-policy PPO backbone would constitute a significant research deviation rather than a direct benchmark of the established method.

Therefore, we utilised Kaleidoscope's original off-policy implementation to ensure a meaningful comparison. We adopted MATD3 as the algorithmic backbone for this evaluation, as it was the only publicly available Kaleidoscope variant with tuned hyperparameters for Multi-Agent MuJoCo (MaMuJoCo). The MaMuJoCo tasks were chosen for alignment with our primary on-policy (IPPO) results and Kaleidoscope's original evaluation, specifically: Ant-v2, HalfCheetah-v2, Walker2d-v2 (overlapping with our IPPO experiments), and Swimmer-v2-10x2 ( which represents the MaMuJoCo variant with the highest number of agents). Comparative results in Table 7 show that HyperMARL achieves competitive results with Kaleidoscope, while only using two critics (standard MATD3) and without additional diversity objectives.

## H.2    Comparison with Selective Parameter Sharing (SePS) - A2C

We benchmark HyperMARL against Selective Parameter Sharing (SePS) [11], a method that shares weights among pre-determined groups of similar agents. SePS identifies these groups by clustering agent trajectories via an autoencoder during a pre-training phase. Following the original authors' setup, we use their provided source code and hyperparameters, evaluating all A2C [32] variants on the Blind Particle Spread (BPS) benchmarks (v1-4), which scale up to 30 agents. For HyperMARL, we sweep the learning rates (1e-4, 3e-4) and hypernetwork hidden dimension (16, 64) per scenario.

As shown in Table 8, HyperMARL demonstrates competitive performance against both NoPS and SePS in these challenging settings, which feature up to 30 agents and five distinct agent roles. Crucially, HyperMARL achieves this result without resorting to agent-specific weights (like NoPS) or requiring a pre-training and clustering pipeline (like SePS).

Table 8: *Mean episode reward in Blind Particle Spread for A2C variants.* We report the IQM of the final total reward (with 95% confidence intervals) across 5 seeds on the BPS environments. **Bold** indicates the highest IQM score; * indicates scores whose 95% confidence intervals overlap with the highest score. HyperMARL achieves results competitive with SePS without requiring a pre-training phase or explicit agent clustering.

| Environment | NoPS | FuPS+ID | SePS | HyperMARL |
|---|---|---|---|---|
| BPS-1 (15 agents, 3 groups, 5–5–5) | -216.8 (-227.7, -197.8) | -228.2 (-247.5, -213.3) | *-201.8 (-221.4, -186.8) | **-190.8 (-220.0, -178.5)** |
| BPS-2 (30 agents, 3 groups, 10–10–10) | *-415.4 (-459.4, -366.3) | -429.7 (-507.1, -411.7) | *-407.1 (-453.7, -387.6) | **-397.8 (-423.7, -376.6)** |
| BPS-3 (30 agents, 5 groups, 6–6–6–6–6) | **-403.4 (-421.0, -381.0)** | -835.2 (-1445.6, -534.1) | *-422.1 (-466.2, -387.6) | *-417.7 (-448.2, -381.1) |
| BPS-4 (30 agents, 5 uneven groups, 2–2–2–15–9) | *-410.8 (-436.6, -368.3) | -780.5 (-1044.1, -593.8) | *-411.6 (-446.6, -346.7) | **-389.5 (-457.6, -366.0)** |

### H.3 Dispersion Detailed Results

### H.3.1 Interval Estimates Dispersion

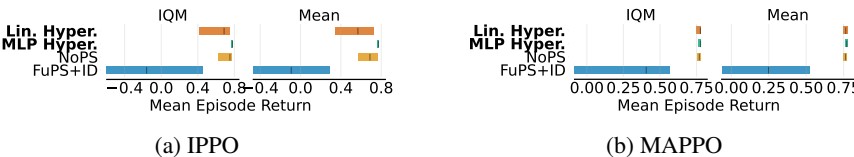

(a) IPPO

(b) MAPPO

Figure 18: *Performance of IPPO and MAPPO on Dispersion after 20 million timesteps.* We show the Interquartile Mean (IQM) of the Mean Episode Return and the 95% Stratified Bootstrap Confidence Intervals (CI) using [1]. Hypernetworks achieve comparable performance to NoPS, while FuPS struggle with specialisation.

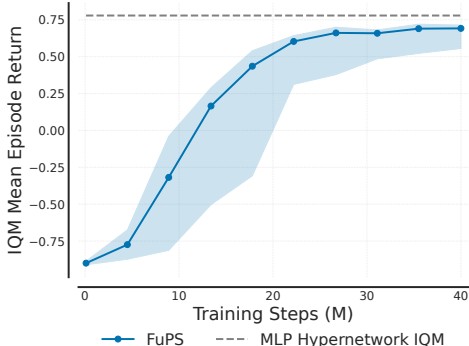

Figure 19: We see that even if we run MAPPO-FuPS on Dispersion for 40 million timesteps (double the timesteps of MLP Hypernetwork), it converges to suboptimal performance.

### H.4 Detailed MAMujoco Plots

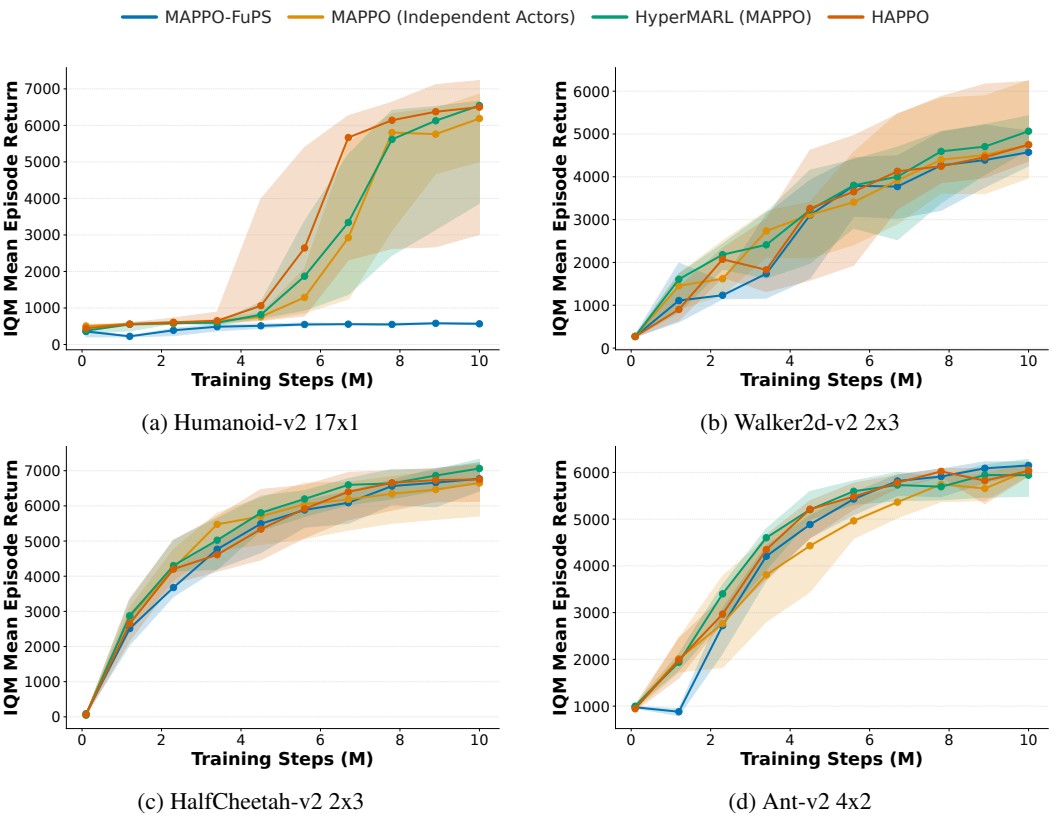

(a) Humanoid-v2 17x1

(b) Walker2d-v2 2x3

(c) HalfCheetah-v2 2x3

(d) Ant-v2 4x2

Figure 20: Performance of Recurrent IPPO and MAPPO on MaMoJoCo. HyperMARL performs comparably to these baselines, and is the only method with shared actors to demonstrate stable learning in the notoriously difficult 17-agent Humanoid environment.

## H.5 Detailed Navigation Plots

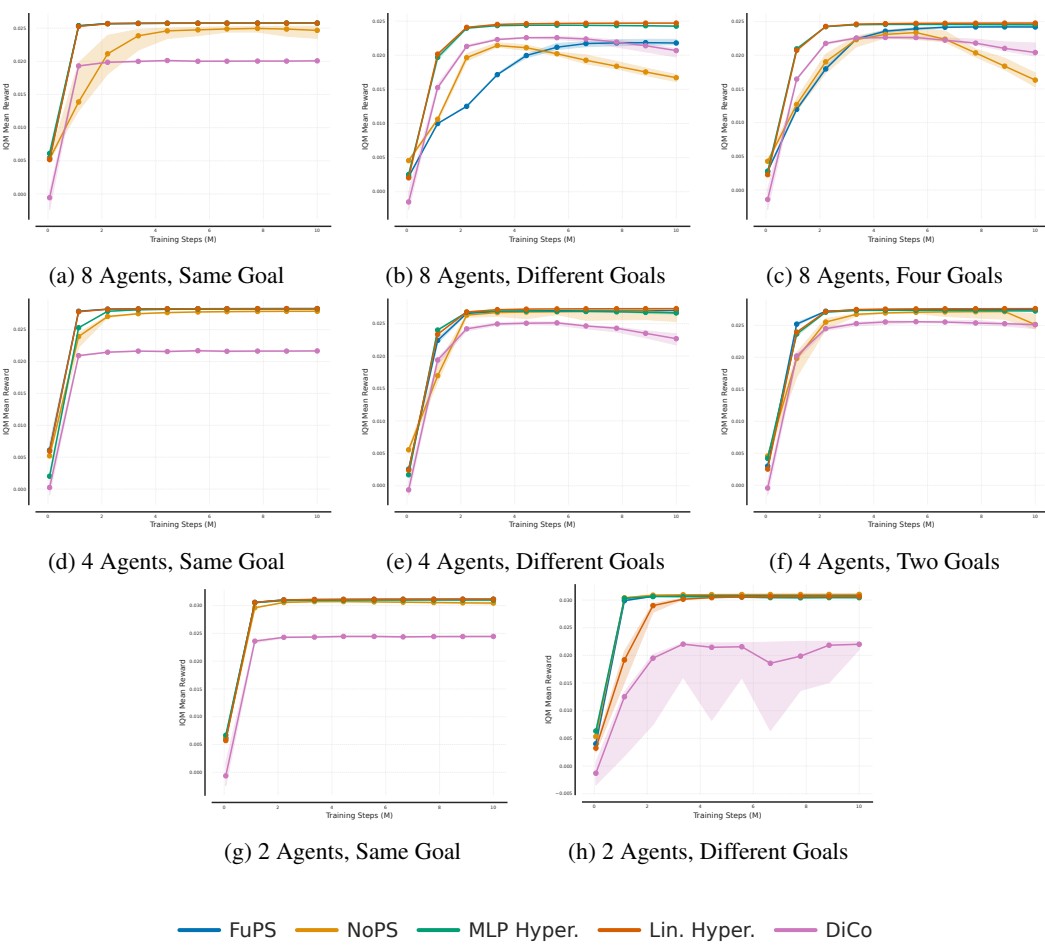

Figure 21: Sample efficiency of IPPO variants in the VMAS Navigation environment. Plots show IQM and 95% CI (shaded regions) of mean episode return against training steps for different agent counts (rows: 8, 4, 2 agents) and goal configurations (columns, where applicable: Same, Different, Specific Goal Counts). Legend shown at bottom applies to all subplots.

## H.6 Interval Estimates - SMAX

## H.7 Additional Ablations

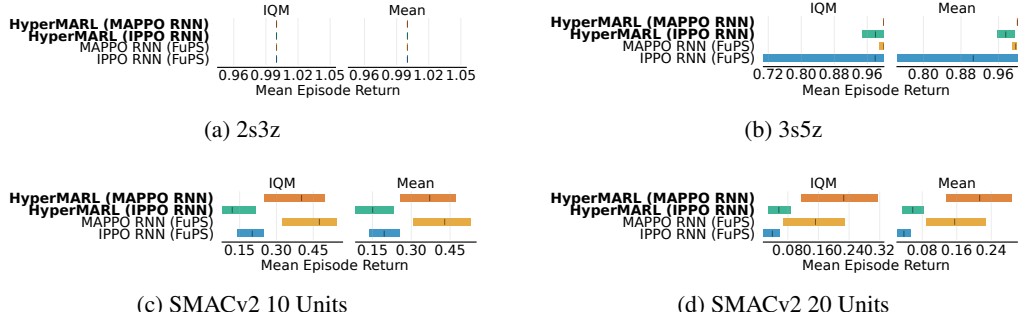

(a) 2s3z

(b) 3s5z

(c) SMACv2 10 Units

(d) SMACv2 20 Units

Figure 22: *Performance comparison in SMAX environments after 10 million timesteps.* We show the Interquartile Mean (IQM) of the Mean Win Rate and the 95% Stratified Bootstrap Confidence Intervals (CI). HyperMARL performs comparably to FuPS baselines across all environments, demonstrating its effectiveness in tasks requiring homogeneous behaviours and using recurrency.

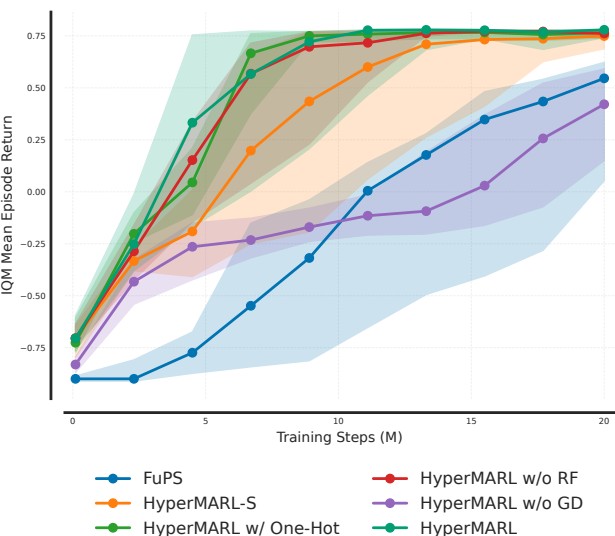

Figure 23: *Ablation results comparing HyperMARL with its variants in Dispersion.* The results highlight that gradient decoupling is essential for maintaining HyperMARL's performance.

# I Hyperparameters

Table 9: Hyperparameters, Training and Evaluation for Specialisation and Synchronisation Game

| Hyperparameter | Value |
|---|---|
| *Environment Configuration* | |
| Number of agents | 2, 4, 8, 16 |
| Maximum steps per episode | 10 |
| *Training Protocol* | |
| Number of seeds | 10 |
| Training steps | 10,000 |
| Evaluation episodes | 100 |
| Evaluation interval | 1,000 steps |
| Batch size | 32 |
| *Model Architecture* | |
| Hidden layer size | 8, 16, 32, 64 |
| Activation function | ReLU |
| Output activation | Softmax |
| *Optimization* | |
| Learning rate | 0.01 |
| Optimizer | SGD |
| *Model Parameter Count* | |
| 2 Agents | NoPS: 60 |
| | FuPS: 58 |
| | FuPS+ID: 74 |
| | FuPS+ID (No State): 42 |
| 4 Agents | NoPS: 352 |
| | FuPS: 240 |
| | FuPS+ID: 404 |
| | FuPS+ID (No State): 148 |
| 8 Agents | NoPS: 2400 |
| | FuPS: 2344 |
| | FuPS+ID: 2600 |
| | FuPS+ID (No State): 552 |
| 16 Agents | NoPS: 17728 |
| | FuPS: 17488 |
| | FuPS+ID: 18512 |
| | FuPS+ID (No State): 2128 |

Table 10: IPPO and MAPPO Hyperparameters in Dispersion

| Hyperparameter | Value |
|---|---|
| LR | 0.0005 |
| GAMMA | 0.99 |
| VF_COEF | 0.5 |
| CLIP_EPS | 0.2 |
| ENT_COEF | 0.01 |
| NUM_ENVS | 16 |
| NUM_STEPS | 128 |
| GAE_LAMBDA | 0.95 |
| NUM_UPDATES | 9765 |
| EVAL_EPISODES | 32 |
| EVAL_INTERVAL | 100000 |
| MAX_GRAD_NORM | 0.5 |
| UPDATE_EPOCHS | 4 |
| NUM_MINIBATCHES | 2 |
| TOTAL_TIMESTEPS | 20000000 |
| ANNEAL_LR | false |
| ACTOR_LAYERS | [64, 64] |
| CRITIC_LAYERS | [64, 64] |
| ACTIVATION | relu |
| SEEDS | 30,1,42,72858,2300658 |
| ACTION_SPACE_TYPE | discrete |

Table 11: MLP Hypernet Hyperparameters in Dispersion

| Parameter | IPPO | MAPPO |
|---|---|---|
| HYPERNET_EMBEDDING_DIM | 4 | 8 |
| EMBEDDING_DIM Sweep | [4, 16, 64] | [4, 8, 16, 64] |
| HYPERNET_HIDDEN_DIMS | 64 | 64 |

Table 12: Dispersion Settings

| Setting | Value |
|---|---|
| n_food | 4 |
| n_agents | 4 |
| max_steps | 100 |
| food_radius | 0.08 |
| share_reward | false |
| penalise_by_time | true |
| continuous_actions | false |

Table 13: IPPO Hyperparameters for Navigation

| Hyperparameters | Value |
|---|---|
| LR | 0.00005 |
| NUM_ENVS | 600 |
| NUM_STEPS | 100 |
| TOTAL_TIMESTEPS | $10^6$ |
| UPDATE_EPOCHS | 45 |
| NUM_MINIBATCHES | 30 |
| GAMMA | 0.9 |
| GAE_LAMBDA | 0.9 |
| CLIP_EPS | 0.2 |
| ENT_COEF | 0.0 |
| VF_COEF | 1.0 |
| MAX_GRAD_NORM | 5 |
| ACTIVATION | tanh |
| ANNEAL_LR | False |
| ACTOR_LAYERS | [256, 256] |
| CRITIC_LAYERS | [256, 256] |
| ACTION_SPACE_TYPE | continuous |

Table 14: MLP Hypernet Hyperparameters in Navigation

| Parameter | IPPO | MAPPO |
|---|---|---|
| HYPERNET_EMBEDDING_DIM | 4 | 8 |
| EMBEDDING_DIM Sweep | [4, 16, 64] | [4, 8, 16, 64] |
| HYPERNET_HIDDEN_DIMS | 64 | 64 |

Table 15: DiCo Algorithm $SND\_des$ Hyperparameter

| Goal Configuration | Number of Agents | SND_des |
|---|---|---|
| All agents same goal | 2 | 0 |
| | 4 | 0 |
| | 8 | 0 |
| All agents different goals | 2 | 1.2 (From DiCo paper) |
| | 4 | [-1,1.2,2.4] $\Rightarrow$ -1 (Best) |
| | 8 | [-1,1.2,4.8] $\Rightarrow$ -1 (Best) |
| Some agents share goals | 4 | [-1,1.2] $\Rightarrow$ -1 (Best) |
| | 8 | [-1,2.4,1.2] $\Rightarrow$ -1 (Best) |

Table 16: Parameter Sweeps for IPPO Variants in Navigation Tasks with Four and Eight Agents

| Parameter Sweeps | |
|---|---|
| CLIP_EPS | 0.2, 0.1 |
| LR | 5e-5, 5e-4, 2.5e-4 |

| Algorithm | Setting | Selected Values |
|---|---|---|
| IPPO-FuPS | 8 Agents (Same Goals) | 0.2, 5e-5 |
| | 8 Agents (Different Goals) | 0.1, 5e-5 |
| | 8 Agents (Four Goals) | 0.1, 5e-5 |
| | 4 Agents (Same Goals) | 0.2, 5e-5 |
| | 4 Agents (Different Goals) | 0.2, 5e-5 |
| | 4 Agents (Two Goals) | 0.2, 5e-5 |
| IPPO-Linear Hypernetwork | 8 Agents (Same Goals) | 0.2, 5e-5 |
| | 8 Agents (Different Goals) | 0.1, 5e-5 |
| | 8 Agents (Four Goals) | 0.1, 5e-5 |
| | 4 Agents (Same Goals) | 0.2, 5e-5 |
| | 4 Agents (Different Goals) | 0.1, 5e-5 |
| | 4 Agents (Two Goals) | 0.1, 5e-5 |
| IPPO-MLP Hypernetwork | 8 Agents (Same Goals) | 0.2, 5e-5 |
| | 8 Agents (Different Goals) | 0.1, 5e-5 |
| | 8 Agents (Four Goals) | 0.1, 5e-5 |
| | 4 Agents (Same Goals) | 0.1, 5e-5 |
| | 4 Agents (Different Goals) | 0.1, 5e-5 |
| | 4 Agents (Two Goals) | 0.1, 5e-5 |
| IPPO-NoPS | 8 Agents (Same Goals) | 0.1, 5e-5 |
| | 8 Agents (Different Goals) | 0.2, 5e-5 |
| | 8 Agents (Four Goals) | 0.1, 5e-5 |
| | 4 Agents (Same Goals) | 0.1, 5e-5 |
| | 4 Agents (Different Goals) | 0.2, 5e-5 |
| | 4 Agents (Two Goals) | 0.1, 5e-5 |
| IPPO-Dico | 8 Agents (Same Goals) | 0.2, 5e-5 |
| | 8 Agents (Different Goals) | 0.1, 2.5e-4 |
| | 8 Agents (Four Goals) | 0.1, 2.5e-4 |
| | 4 Agents (Same Goals) | 0.2, 5e-5 |
| | 4 Agents (Different Goals) | 0.1, 2.5e-4 |
| | 4 Agents (Two Goals) | 0.1, 5e-4 |

Table 17: Environment Settings for Navigation Task

| Parameter | Value |
|---|---|
| n_agents | 2,4,8 |
| agents_with_same_goal | 1, n_agents/2, n_agents |
| max_steps | 100 |
| collisions | False |
| split_goals | False |
| observe_all_goals | True |
| shared_rew | False |
| lidar_range | 0.35 |
| agent_radius | 0.1 |
| continuous_actions | True |

Table 18: Default algorithm and model hyperparameters for the Ant-v2-4x2 environment (from [54]).

| Parameter | Value |
|---|---|
| **— Algorithm Parameters —** | |
| action_aggregation | prod |
| actor_num_mini_batch | 1 |
| clip_param | 0.1 |
| critic_epoch | 5 |
| critic_num_mini_batch | 1 |
| entropy_coef | 0 |
| fixed_order | true |
| gae_lambda | 0.95 |
| gamma | 0.99 |
| huber_delta | 10.0 |
| max_grad_norm | 10.0 |
| ppo_epoch | 5 |
| share_param | false |
| use_clipped_value_loss | true |
| use_gae | true |
| use_huber_loss | true |
| use_max_grad_norm | true |
| use_policy_active_masks | true |
| value_loss_coef | 1 |
| **— Model Parameters —** | |
| activation_func | relu |
| critic_lr | 0.0005 |
| data_chunk_length | 10 |
| gain | 0.01 |
| hidden_sizes | [128, 128, 128] |
| initialization_method | orthogonal_ |
| lr | 0.0005 |
| opti_eps | 1e-05 |
| recurrent_n | 1 |
| std_x_coef | 1 |
| std_y_coef | 0.5 |
| use_feature_normalization | true |
| use_naive_recurrent_policy | false |
| use_recurrent_policy | false |
| weight_decay | 0 |

Table 19: Default algorithm and model hyperparameters for the Humanoid-v2-17x1 environment (from [54]).

| Parameter | Value |
|---|---|
| **— Algorithm Parameters —** | |
| action_aggregation | prod |
| actor_num_mini_batch | 1 |
| clip_param | 0.1 |
| critic_epoch | 5 |
| critic_num_mini_batch | 1 |
| entropy_coef | 0 |
| fixed_order | false |
| gae_lambda | 0.95 |
| gamma | 0.99 |
| huber_delta | 10.0 |
| max_grad_norm | 10.0 |
| ppo_epoch | 5 |
| share_param | false |
| use_clipped_value_loss | true |
| use_gae | true |
| use_huber_loss | true |
| use_max_grad_norm | true |
| use_policy_active_masks | true |
| value_loss_coef | 1 |
| **— Model Parameters —** | |
| activation_func | relu |
| critic_lr | 0.0005 |
| data_chunk_length | 10 |
| gain | 0.01 |
| hidden_sizes | [128, 128, 128] |
| initialization_method | orthogonal_ |
| lr | 0.0005 |
| opti_eps | 1e-05 |
| recurrent_n | 1 |
| std_x_coef | 1 |
| std_y_coef | 0.5 |
| use_feature_normalization | true |
| use_naive_recurrent_policy | false |
| use_recurrent_policy | false |
| weight_decay | 0 |

Table 20: Default algorithm and model hyperparameters for the Walker2d-v2-2x3 environment (from [54]).

| Parameter | Value |
|---|---|
| **— Algorithm Parameters —** | |
| action_aggregation | prod |
| actor_num_mini_batch | 2 |
| clip_param | 0.05 |
| critic_epoch | 5 |
| critic_num_mini_batch | 2 |
| entropy_coef | 0 |
| fixed_order | false |
| gae_lambda | 0.95 |
| gamma | 0.99 |
| huber_delta | 10.0 |
| max_grad_norm | 10.0 |
| ppo_epoch | 5 |
| share_param | false |
| use_clipped_value_loss | true |
| use_gae | true |
| use_huber_loss | true |
| use_max_grad_norm | true |
| use_policy_active_masks | true |
| value_loss_coef | 1 |
| **— Model Parameters —** | |
| activation_func | relu |
| critic_lr | 0.001 |
| data_chunk_length | 10 |
| gain | 0.01 |
| hidden_sizes | 128, 128, 128 |
| initialization_method | orthogonal_ |
| lr | 0.001 |
| opti_eps | 1e-05 |
| recurrent_n | 1 |
| std_x_coef | 1 |
| std_y_coef | 0.5 |
| use_feature_normalization | true |
| use_naive_recurrent_policy | false |
| use_recurrent_policy | false |
| weight_decay | 0 |

Table 21: Default algorithm and model hyperparameters for the HalfCheetah-v2-2x3 environment (from [54]).

| Parameter | Value |
|---|---|
| **— Algorithm Parameters —** | |
| action_aggregation | prod |
| actor_num_mini_batch | 1 |
| clip_param | 0.05 |
| critic_epoch | 15 |
| critic_num_mini_batch | 1 |
| entropy_coef | 0.01 |
| fixed_order | false |
| gae_lambda | 0.95 |
| gamma | 0.99 |
| huber_delta | 10.0 |
| max_grad_norm | 10.0 |
| ppo_epoch | 15 |
| share_param | false |
| use_clipped_value_loss | true |
| use_gae | true |
| use_huber_loss | true |
| use_max_grad_norm | true |
| use_policy_active_masks | true |
| value_loss_coef | 1 |
| **— Model Parameters —** | |
| activation_func | relu |
| critic_lr | 0.0005 |
| data_chunk_length | 10 |
| gain | 0.01 |
| hidden_sizes | 128, 128, 128 |
| initialization_method | orthogonal_ |
| lr | 0.0005 |
| opti_eps | 1e-05 |
| recurrent_n | 1 |
| std_x_coef | 1 |
| std_y_coef | 0.5 |
| use_feature_normalization | true |
| use_naive_recurrent_policy | false |
| use_recurrent_policy | false |
| weight_decay | 0 |

Table 22: HyperMARL Hyperparameters Across MaMuJoCo Environments

| Parameter | Humanoid v2-17x1 | Walker2d v2-2x3 | HalfCheetah v2-2x3 | Ant v2-4x2 | Sweeps |
|---|---|---|---|---|---|
| AGENT_ID_DIM | 64 | 64 | 64 | 64 | None |
| HNET_HIDDEN_DIMS | 64 | 64 | 64 | 64 | None |
| clip_param | 0.075 | 0.0375 | 0.0375 | 0.075 | [0.1,0.075,0.05,0.0375] |

Table 23: Recurrent IPPO and MAPPO Hyperparameters in SMAX (from JaxMARL paper)

| Hyperparameter | IPPO Value | MAPPO Value |
|---|---|---|
| LR | 0.004 | 0.002 |
| NUM_ENVS | 128 | 128 |
| NUM_STEPS | 128 | 128 |
| GRU_HIDDEN_DIM | 128 | 128 |
| FC_DIM_SIZE | 128 | 128 |
| TOTAL_TIMESTEPS | 1e7 | 1e7 |
| UPDATE_EPOCHS | 4 | 4 |
| NUM_MINIBATCHES | 4 | 4 |
| GAMMA | 0.99 | 0.99 |
| GAE_LAMBDA | 0.95 | 0.95 |
| CLIP_EPS | 0.05 | 0.2 |
| SCALE_CLIP_EPS | False | False |
| ENT_COEF | 0.01 | 0.0 |
| VF_COEF | 0.5 | 0.5 |
| MAX_GRAD_NORM | 0.25 | 0.25 |
| ACTIVATION | relu | relu |
| SEED | 30,1,42,72858,2300658 | 30,1,42,72858,2300658 |
| ANNEAL_LR | True | True |
| OBS_WITH_AGENT_ID | - | True |

Table 24: Hyperparameter Sweeps and Final Values for Different Maps in SMAX. H- refers to HyperMARL MLP Hypernetworks.

| Map | Algorithm | LR Range | Chosen LR | HNET Embedding Dim | HNET Hidden Dims |
|---|---|---|---|---|---|
| **2s3z** | IPPO | 0.004 | 0.004 | – | |
| | MAPPO | 0.002 | 0.002 | – | |
| | H-IPPO | 0.004 | 0.004 | 4 | 32 |
| | H-MAPPO | 0.002 | 0.002 | 64 | 16 |
| **3s5z** | IPPO | 0.004 | 0.004 | – | |
| | MAPPO | 0.002, 0.005, 0.0003 | 0.002 | – | |
| | H-IPPO | 0.004 | 0.004 | 64 | 16 |
| | H-MAPPO | 0.002, 0.005, 0.0003 | 0.0003 | 64 | 16 |
| **smacv2_10_units** | IPPO | 0.005, 0.001, 0.0003, 0.004 | 0.001 | – | |
| | MAPPO | 0.002, 0.005, 0.0003 | 0.0003 | – | |
| | H-IPPO | 0.005, 0.001, 0.0003, 0.004 | 0.005 | 4 | 64 |
| | H-MAPPO | 0.002, 0.005, 0.0003 | 0.0003 | 64 | 16 |
| **smacv2_20_units** | IPPO | 0.002, 0.005, 0.0003 | 0.005 | – | |
| | MAPPO | 0.002, 0.005, 0.0003 | 0.0003 | – | |
| | H-IPPO | 0.002, 0.005, 0.0003 | 0.005 | 64 | 64 |
| | H-MAPPO | 0.002, 0.005, 0.0003 | 0.0003 | 4 | 64 |

*Note:* HNET Embedding Dim refers to the hypernetwork embedding dimension, chosen from the range {4, 16, 64}. HNET Hidden Dims refers to the hidden layer dimensions of the hypernetwork, chosen from the range {16, 32, 64}.

## J   Computational Resources

Table 25: Computational Resources by Experiment Type

| Experiment Category | Hardware Configuration | Execution Time | Total Hours |
|---|---|---|---|
| Specialisation, Synchronisation & Dispersion | 8 cores on AMD EPYC 7H12 64-Core Processor | 2-6 hours per run (agent-count dependent) | 250 |
| Navigation Experiments | 8 cores on AMD EPYC 7H12 64-Core Processor + NVIDIA RTX A4500 | 4-10 hours per run | 320 |
| MaMuJoCo Experiments | 8 cores on AMD EPYC 7H12 64-Core Processor + NVIDIA RTX A4500 | 8-24 hours per run (scenario & algorithm dependent) | 1,680 |
| SMAX Experiments | 8 cores on AMD EPYC 7H12 64-Core Processor + NVIDIA RTX A4500 | 2-5 hours per run | 160 |

