# OpenReview forum: "HyperMARL: Adaptive Hypernetworks for Multi-Agent RL"
_NeurIPS.cc/2025/Conference — NeurIPS 2025 poster_

### Official Review · Reviewer_QQ2N · 2025-06-13

**Clarity:** 3
**Significance:** 2
**Originality:** 2
**Rating:** 5
**Confidence:** 4

**Summary:**

In this work, the authors focus on the potential limitations in parameter sharing methods. Based on experimental evidence, it is elaborated that coupling agent IDs with observations can be detrimental, blamed by gradient interference among agents. To overcome the key difficulty with FuPS, an agent-conditioned hypernetworks based parameter sharing method is proposed, which explicitly decouples observation- and agent-conditioned gradients. By mitigating cross-agent interference, HyperMARL empirically reduces policy gradient variance, facilitates shared-policy adaptation, and learns heterogeneous, homogeneous, or mixed behaviors, as demonstrated in various challenging MARL benchmarks.

**Questions:**

1. Is it possible that the specialization of the policy may lead to redundant diversification policies? For example, in a neural network, a function can be fitted with two nodes, or it can be fitted with MLP, but the latter may cause serious overfitting problems. Will the strengthening of specialization lead to a decrease in adaptability? How to know the appropriate level of specialization is achieved?

2. (Christianos, Filippos, et al. "Scaling multi-agent reinforcement learning with selective parameter sharing." International Conference on Machine Learning. PMLR, 2021). The above paper seems to have similar ideas, and their encoder-decoder can also be considered as a type of hypernetworks. What are the similarities and differences between this work and previous work? Furthermore, in the previous work, the encoder-decoder was trained before RL, while in this paper, it is trained simultaneously. During simultaneous training, will there be conflicts between RL learning and policy specialization learning? What are the advantages of simultaneous training? Based on the decoupling idea mentioned in this article, it seems that it would be better to decouple these two optimization goals?

3. In Figure 3 (page 4), why ‘FuPS+ID’ method fails even in synchronization environment? According to the explanation in subsection 3.2, it should not fail in this environment? In Figure 4, what is ‘PG-FuPS+ID’ (with no definition)? Is ‘PG-FuPS+ID’ equal to ‘FuPS+ID’? In line 105 (page 4), ‘Table 3’ (mistake?)? In line 108 to 109, why authors claim that the fully policy sharing method also has scalability issue in synchronization environment?

4. In subsection 3.2 (line 117 ‘Importance of Observation and ID Decoupling’), authors show the result of ‘FuPS+ID’ and ‘FuPS+ID (No State)’. What about ‘FuPS’? Is it an orthogonal gradient or a more conflicting gradient? If it is orthogonal gradient: why is it not performing well? If not: is it possible that adding ID could reduce the conflicting gradients, that is, ID is actually beneficial? Based on the existing results, it seems that we cannot easily draw a conclusion about "Importance of Observation and ID Decoupling"?

5. In section 4, how are the agent-specific parameters ($\theta $) integrated into original RL networks? Are all the parameters of the policy network ($\pi $) generated by the hypernetworks? Or is it just the last layer that is replaced, and the other layers are still RNN? Why use ID as input to the hypernetworks? Can the inputs to the hypernetworks and policy network be swapped (i.e. use observation as input to the hypernetworks, since the purpose here seems to be just for decoupling)?

**Ethical Concerns:**

["NO or VERY MINOR ethics concerns only"]

**Final Justification:**

Based on the rebuttal and subsequent discussion, I believe the authors have addressed most of my concerns. I would like to confirm my score as 5 and recommend accepting.

**Limitations:**

Yes

**Paper Formatting Concerns:**

N.A.

**Quality:**

3

**Strengths And Weaknesses:**

Strengths:

1. The paper is well-written. The organization of this paper is clear, and the proposed idea is easy to follow. The limitations of parameter sharing are elaborated with clear definition of research problems. The experimental verification on the impact of the coupling of observations and agent IDs brings certain level of new opinion and knowledge.

2. The experiment is comprehensive. The proposed method is tested in diverse benchmarks where heterogeneous, homogeneous, or mixed behaviors are required. The ablation and computation complexity are also considered.

3. Even though simple, the methodology is effective. The theory behind has been carefully discussed with verified hypernetwork gradients. The key idea of decoupling is well introduced and proved.

Weaknesses:

1. Some discussion of related work seems to be missing. Part of this work seems similar to “Scaling Multi-Agent Reinforcement Learning with Selective Parameter Sharing”. Maybe adding further discussion in the related work section could be helpful.

2. In methodology section (Section 4), how the agent-specific parameters ($\theta $) are integrated into original RL networks seems not to be clear.

3. The argument in Section 3 seems unsatisfactory. There seems to be some small mistakes.

Quality: 2.9 $\approx $ 3 (From introduction to conclusion section, the discussion of parameter sharing is comprehensive.);

Clarity: 3.3 $\approx $ 3 (Overall, this paper is well-written with clear definition of research problem. It is easy to follow the idea in this paper.);

Significance: 2.4 $\approx $ 2 (Parameter sharing is a well-discussed topic. It seems that part of this work is considered by previous work?);

Originality: 2.4 $\approx $ 2 (I am more likely to view this paper as an extension of previous parameter sharing work?);

Rating: 4.5 $\approx $ 5 (Overall, this is an interesting paper. If authors can address my concerns, I will confirm the score to 5.);

Confidence: 3.6 $\approx $ 4 (The paper is good and easy to follow. But I am not quite sure about the key difference between it and previous work.).

---

> ### Author Rebuttal · Authors · 2025-07-31
>
> We thank reviewer `QQ2N` for their thorough and thoughtful review. We appreciate the reviewer's positive feedback on our novel insight into gradient decoupling, our comprehensive experimental evaluation, and our paper's clarity. We address your points as follows:
>
> ## **1. SePS[1] - Related work (W1, Q2)**
>
> We thank the reviewer for this point. SePS [1] is a parameter sharing method that uses an autoencoder during *pre-training* to cluster agents based on their transitions and shares parameters between these agent groups. This differs from our work, as we 1) do not use a *pre-training* phase to collect transitions and cluster agents, 2) do not maintain $K$ per-agent networks, but rather generate these networks on the fly based on agent embeddings, and 3) we use agent-conditioned hypernetworks that explicitly decouple observation- and agent-conditioned gradients (Section 4.3). This decoupling helps mitigate gradient interference between agents. Although we briefly mentioned SePS in our related work, we will improve our discussion of this work.
>
> **New Baseline and Environment:** Furthermore, since this work was mentioned as closely related by two reviewers, we conducted additional experiments comparing HyperMARL to SePS. We use their original source code and hyperparameters and evaluate on all their Blind Particle Spread settings (v1-4), which include up to **30 agents**.
>
> IQM final total reward (with 95% confidence intervals) across 5 seeds on the BPS environments from [1], Bold indicates the highest IQM score; * indicates scores whose 95% confidence intervals overlap with the highest score.
>
> | Alg | BPS-1 (15 agents, 3 groups)  | BPS-2 (30 agents, 3 groups)  | BPS-3 (30 agents, 5 groups) | BPS-4 (30 agents, 5 uneven groups) |
> |-----------|----|----|----|----|
> | NOPS | **-205.4** (-242.2, -197.8) | **-405.1** (-468.5, -379.9) | *-420.3 (-439.9, -407.4) | **-413.5** (-449.8, -399.1) |
> | FUPS-ID | *-219.8 (-249.7, -207.1) | *-442.6 (-467.4, -383.7) | -764.7 (-1326.7, -515.3) | -705.9 (-933.3, -591.1) |
> | SEPS | *-216.3 (-233.5, -191.8) | *-413.0 (-464.9, -385.1) | **-403.1** (-423.7, -352.9) | *-418.3 (-436.0, -393.5) |
> | HyperMARL | *-205.6 (-223.3, -175.1) | *-421.0 (-436.7, -401.0) | *-414.0 (-442.8, -374.2)  | *-428.5 (-462.7, -376.3) |
>
> **Takeaway**: The results highlight HyperMARL's competitive performance. In all four scenarios, its 95% confidence intervals overlap with SePS and NoPS, indicating it is statistically on par with these baselines. Notably, SePS relies on a **pre-training clustering phase**, whereas HyperMARL learns end-to-end without pre-training.
>
> ## 2. **Agent-Specific Parameters Integrated into the Agent’s Networks (W2, Q5)**
>
> A hypernetwork generates the agent-specific parameters $\theta^i$ that form all the weights (and biases) of policy $\pi_{\theta^i}$ (can also be seen in Fig. 1).
>
> For RNNs,  HyperMARL maintains the same architecture as for feedforward methods, generating only the actor and critic feedforward weights, not the GRU weights (we also mention this in G.2, but we will highlight this in the main text). The RNN hidden state is then fed into the networks as input, as is common in recurrent architectures in RL.
>
> ## 3. **Could specialisation lead to redundant diversification policies? Could specialisation lead to a decrease in adaptability? (Q1)**
>
> Thank you for these insightful questions. Our opinion is that enforced specialisation, for example enforcing a diversity loss/objective, could lead to redundant diversification when not necessary to solve a task, since the mutual information objective could simply enforce different agent policies to be different, irrespective of the task. This could also hamper adaptability to tasks that do not require specialisation. This is the exact reason we do not *enforce specialisation*, but rather *enable* it, by learning end-to-end when specialised or homogeneous behaviours are required. To validate this, we experimentally test our approach in a wide array of settings, requiring homogeneous, heterogeneous, or mixed behaviours.
>
> ## 4. **FuPS+ID in the Synchronisation Environment (Q4)**
>
> FuPS+ID performs well in the Synchronisation environment with a few agents, but as the number of agents increases, this environment becomes more challenging and FuPS+ID performance drops. This is because the probability of getting the maximum reward (all agents selecting the same action) during exploration is $\frac{n!}{n^n}$, where $n$ is the number of agents e.g. $n=8$, $Pr(X = max_reward)=0.002$. This means that as $n$ grows large, it is harder for agents to synchronise. Agents get rewards for partial matches as described in section E.1 in the Appendix, but this is still a challenging environment. We will make this clearer in the updated manuscript.
>
> ## 5. **FuPS in the Gradient Conflict Section (3.2) (Q5)**
>
> The primary goal of Section 3.2 was to investigate why FuPS+ID struggles to specialise, despite being a universal approximator ($\pi_{\theta}(a^i|o^i,id^i)$ should be able to approximate $\pi_{{\theta}_i}$). As you note, our analysis in this section omits the plain FuPS baseline because it has a more fundamental limitation: a shared policy without agent IDs cannot learn distinct agent behaviours. We prove this formally for the 2-player Specialisation Game in Appendix E.3. The proof and empirical results shown in Table 3 show that agent IDs can be beneficial.
>
> However, our central claim is that the method of providing this ID is critical. The FuPS+ID (No State) ablation, which outperforms standard FuPS+ID by ignoring observations, demonstrates that naively coupling the observation and ID inputs introduces destructive gradient interference. Therefore, the conclusion about the "Importance of Observation and ID Decoupling" still holds -- while IDs are necessary for specialisation, they must be decoupled from observation processing to be most effective. We will clarify this in the paper.
>
> ## 6. **Can the inputs to the hypernetworks and policy network be swapped (i.e. use observation as input to the hypernetworks, …) (Q5)?**
>
> This is an interesting idea, but we believe this will likely not work. Swapping the inputs would re‑entangle ID and observations. The agent‑conditioned Jacobian in Eq. (4) would become observation‑dependent (not deterministic with respect to mini-batch samples), removing the variance/decoupling benefit that HyperMARL relies on. Furthermore, this would result in other challenges as an entirely new policy would be generated every timestep.  The agents would also effectively be blind since the policy is not conditioned on observations, and therefore would likely not perform well in most environments (even if the weights were generated conditioned on the observations).
>
> ## 7. **Minor Wording (Q3)**
> - "PG-FuPS+ID" - Thanks for pointing this out. Apologies, "PG-FuPS+ID" is the same as "FuPS+ID"; PG stands for policy gradient. We will fix this minor wording error in the updated manuscript.
> - Lines 108-109 - Thanks for the clarifying question. Our claim regarding "scalability" refers not to FuPS's performance degradation, but to the widening performance gap between the FuPS and NoPS architectures as the number of agents increases, i.e. for larger values of $n$, the gap between FuPS and NoPS increases (NoPS performing better in Specialisation, FuPS in Synchronisation). We will clarify the wording accordingly.
>
>
> ## **Conclusion**
>
> In summary, based on your feedback we added:
> - A **new baseline** SePS and environment, Blind Particle Spread. Extending our evaluation to **22 scenarios, up to 30 agents and six baselines** (NoPS, FuPS, DiCo, HAPPO, Kaleidoscope and now SePS). HyperMARL is competitive with this baseline.
> - Clarified the integration of agent-specific parameters into RL networks, the interplay between specialisation and redundant diversification, and other clarifications.
> - Clarified the FuPS results in the Synchronisation Environment and the Gradient Conflict Section.
>
> During the rebuttal, we have also added:
> - A **sensitivity analysis** using recurrent MAPPO in the challenging 20-agent SMAC-v2 style environment.
> - Discussed the computational trade-off of using our method and referred to our existing sections on the memory, speed and parameter efficiency of our method (sections F3, F4).
> - A **learned agent embeddings analysis** showing the embeddings move closer together when the agents should behave similarly, and embeddings remain far apart when agents need to specialise.
> - A comparison with **dual-channel networks**, showing our approach is more effective.
>
> We would like to thank the reviewer again for their thorough review and questions. We hope these clarifications and additional experiments have fully addressed the points raised, and if you have any follow-up questions or comments, please let us know, and we will be happy to discuss further.
>
>
> References:
>
> 1] Christianos, F., Papoudakis, G., Rahman, M.A. and Albrecht, S.V., 2021, July. Scaling multi-agent reinforcement learning with selective parameter sharing. In International Conference on Machine Learning (pp. 1989-1998). PMLR.
>
> 2] Kurt Hornik, Maxwell Stinchcombe, and Halbert White. Multilayer feedforward networks are universal approximators. Neural networks, 2(5):359–366, 1989.

---

> > ### Comment · Reviewer_QQ2N · 2025-08-02
> >
> > Thanks for the comprehensive explanation. I appreciate the newly conducted experiments and further clarification, which address most of my concerns (fully or partially). Although I was hoping to see a clear result for FuPS’s ‘Avg. gradient conflict’, the current explanation still works for me. All in all, the proof and analysis provided in the authors’ rebuttal reinforce their claims and contributions. Even though I continue to view this work as an extension of previous work (such as ‘SePS’), it offers valuable theoretical and experimental insights that should benefit the community. This is a good paper, and I would like to confirm my score as 5 (4.5 $\to $ 4.8).
> >
> > By the way, does the ‘Table 3’ you mentioned in “5. FuPS in the Gradient Conflict Section (3.2) (Q5)” refer to ‘Table 3: Measure Policy Diversity’ in the appendix of the paper (Page 15)?

---

> > > ### Author Response · Authors · 2025-08-02
> > > **Thank you!**
> > >
> > > We thank reviewer `QQ2N` for the continued engagement and for confirming your positive assessment. Your detailed and thorough feedback has been instrumental in strengthening the paper.
> > >
> > > Regarding your question, our reference to 'Table 3' was a typographical error in the rebuttal text. We intended to refer to Figure 3 (Average evaluation reward) on page 4. Those results, specifically comparing FuPS and FuPS+ID in the Specialisation Game, were meant to highlight that incorporating agent IDs can be beneficial, which motivates our subsequent analysis of *how* they are incorporated.
> > >
> > > Thank you again for your constructive review. We would be happy to address any other questions.

---

### Official Review · Reviewer_z98m · 2025-06-26

**Clarity:** 4
**Significance:** 3
**Originality:** 2
**Rating:** 4
**Confidence:** 3

**Summary:**

The paper proposes HyperMARL, which decouples the observation-conditioned and agent ID-conditioned gradients via hypernetworks to enable better coordination for both specialist and generalist behaviors. The authors first show that full parameter sharing leads to conflicting updating gradients of agents. Then they propose HyperMARL that uses agent ID-conditioned hypernetworks to generate weights and biases for agent policy and critic networks. Experiments on Dispersion, Navigation, MAMuJoCo, and SMAX validate the proposed HyperMARL when compared with various baselines.

**Questions:**

1.	Could the authors compare hypernetworks with a dual-channel network, where one channel is for agent observation and one channel is for agent ID?
2.	Could HyperMARL work well in complicated tasks such as SMACv2 where the observation provides key decision information? May hypernetworks limit the learning ability of the agent network?
3.	Does HyperMARL support value-factorization MARL algorithms such as QMIX and QPLEX?

**Ethical Concerns:**

["NO or VERY MINOR ethics concerns only"]

**Final Justification:**

Although I recommend accepting, this paper could benefit from comparing with role-based approaches and conducting experiments in SMAC.

**Limitations:**

yes

**Quality:**

3

**Strengths And Weaknesses:**

Strengths

1.	The finding of conflicting gradients among agents by full parameter sharing and agent ID is important.
2.	The method is well-motivated and reasonable. Through the hypernetwork, the gradient flow is explicitly decoupled as shown in Equation (4).
3.	This paper is well-written and well-organized.

Weaknesses

1.	Using agent ID or embedding only to generate weights and biases of agent policy and critic networks may limit the representation ability in complex tasks.
2.	The widely used SMAC and SMACv2 environments are not tested. For the SMAX, its dynamic seems to be simplified and ease the learning of hypernetworks for policy and critic weights.
3.	Some role-based MARL works, such as RODE, are not compared in the experiments.

---

> ### Author Rebuttal · Authors · 2025-07-31
>
> We thank reviewer `z98m` for their review and for encouraging comments regarding our novel insight with respect to gradient decoupling in MARL, our strong motivation for using agent-conditioned hypernetworks and our clarity. We address your comments as follows:
>
> ## 1. **Using ID or Embeddings … may limit representation ability (W1)**
>
> We would like to clarify that the agent IDs or embeddings inputs are not the main components of the hypernetworks that influence their representation ability, but rather the functional form of the hypernetwork. Linear hypernetworks can only capture linear relationships between agents’ embeddings and their corresponding weights. To enhance expressiveness, we introduced MLP hypernetworks for most of our experiments, allowing our method to adapt to more complex tasks. Furthermore, using learned embeddings means that the conditioning can automatically adapt to the task. We will make this property clearer in the updated paper.
>
> ## 2. **Using SMAC and SMACv2 vs SMAX and Could HyperMARL work well in... SMACv2 (W2, Q2)**
>
> We evaluate on SMAX, an accelerated approximate reimplementation of SMAC, which includes v1 and v2 style scenarios. We benchmark on both v1 (e.g. 2s3z,3s5z) and v2 style maps from SMAX (10 and 20 units). Although some dynamics are different in SMAX, it features a more sophisticated heuristic AI opponent, and some maps are harder in this version [1]. We believe these settings provide a strong and appropriate testbed for our method.
>
> Furthermore, we included SMACv2-style scenarios in SMAX that also have this “meaningful partial observability” property of SMACv2-style maps. Our approach performed well in this case (Fig. 8c and d) showing that the hypernetworks do not limit the agent's network.
>
> ## 3. **Comparison with role-based approaches, such as ROMA, in the Experiments (W3)**
>
> Role-based methods, like ROMA and RODE [3, 4], assign agents to subtasks/roles but rely on mutual information (MI) objectives for diversity. Our primary reason for not including these methods in the experiments is that they have been shown to have limited empirical performance (Table 1 in [2]), which we discussed in Table 4 in the Appendix. Furthermore, we have six baselines (NoPS, FuPS, DiCo, HAPPO, Kaleidoscope and now SePS), which we feel serve as a good testbed. Nonetheless, we will expand on our discussion of role-based methods in the related work, in the updated manuscript.
>
> ## 4. **Comparison with Dual-channel Network Architecture**
>
> **New Ablation:** We thank the reviewer for their thoughtful suggestion. Following the reviewer's advice, we implemented and evaluated a **dual-channel network** architecture. This baseline processes the agent's observation and ID in separate channels before concatenating their outputs. We tested two variants: one using one-hot encoded agent IDs and another using learned agent embeddings. We evaluated this using IPPO on the Dispersion task from our paper. Our results below show that HyperMARL outperforms dual-channel networks.
>
> **Results:** IQM and 95% CI of mean test episode return across 5 seeds, **Bold** indicates the highest IQM score.
>
> | Method | IQM, 95% CI |
> |---|---|
> |  HyperMARL (MLP hypernetwork) | **0.7783** (0.7768, 0.7802) |
> |  Dual-Channel (one-hot ID) | -0.4749 (-0.9187, 0.3984) |
> |  Dual -Channel (with Agent Embedding) | -0.2934 (-0.9187, 0.5450) |
>
> **Takeaway:** HyperMARL outperforms dual-channel architectures.
>
> ## 5. **Compatibility with value-factorisation/decomposition methods (e.g. QMIX, QPLEX) (Q3)**
>
> Yes, HyperMARL is compatible with these methods as it simply generates policy or critic networks and could do so in any setting or method. In Table 5 in the App, we showed that HyperMARL also works well in off-policy methods such as MATD3. We believe these results would also hold for value-decomposition methods. However, due to limited time resources and time during the rebuttal, we are unable to run additional experiments using HyperMARL for value-decomposition methods. We will discuss this as a possible future work direction in the updated manuscript.
>
>
> ## **Conclusion**
>
> In summary, based on your feedback we added:
> - A comparison with **dual-channel networks**, showing our approach is more effective.
> - Discussed the compatibility of our method with value-decomposition methods and discussed the representations that can be learned.
>
>
> During the rebuttal, we have also added:
> - A **new baseline** SePS and environment, Blind Particle Spread. Extending our evaluation to **22 scenarios, up to 30 agents and six baselines** (NoPS, FuPS, DiCo, HAPPO, Kaleidoscope and now SePS). HyperMARL is competitive with this baseline.
> - A **sensivitiy analysis** using recurrent MAPPO in the challenging 20-agent SMAC-v2 style environment.
> - A **learned agent embeddings analysis** showing the embeddings move closer together when the agents should behave similarly, and embeddings remain far apart when agents need to specialise.
>
> We would like to thank the reviewer again for their thoughtful review and hope these clarifications and additional experiments address your concerns. If there are no further issues, we respectfully ask reviewer `z98m` to provide an updated assessment of our work.
>
> References:
>
> 1] Rutherford, A., Ellis, B., Gallici, M., Cook, J., Lupu, A., Ingvarsson Juto, G., Willi, T., Hammond, R., Khan, A., Schroeder de Witt, C. and Souly, A., 2024. Jaxmarl: Multi-agent rl environments and algorithms in jax. Advances in Neural Information Processing Systems, 37, pp.50925-50951.
>
> 2] Filippos Christianos, Lukas Schäfer, and Stefano V. Albrecht. Shared experience actor-critic for multi-agent
> reinforcement learning. In 34th Conference on Neural Information Processing Systems, 2020.
>
> 3] Tonghan Wang, Heng Dong, Victor Lesser, and Chongjie Zhang. Roma: multi-agent reinforcement learning with emergent roles. In Proceedings of the 37th International Conference on Machine Learning, 407 pages 9876–9886, 2020.
>
> 4] Tonghan Wang, Tarun Gupta, Anuj Mahajan, Bei Peng, Shimon Whiteson, and Chongjie Zhang. Rode: Learning roles to decompose multi-agent tasks. arXiv preprint arXiv:2010.01523, 2020.

---

> > ### Comment · Reviewer_z98m · 2025-08-01
> >
> > Thanks to the authors for their responses. Most of my concerns are (partially) addressed, and I would like to maintain my score.

---

### Official Review · Reviewer_uZHN · 2025-06-30

**Clarity:** 4
**Significance:** 2
**Originality:** 2
**Rating:** 4
**Confidence:** 3

**Summary:**

This paper addresses a central challenge in cooperative multi-agent reinforcement learning (MARL): enabling policies to adaptively learn specialized, homogeneous, or mixed behaviors without sacrificing sample efficiency. The authors identify an issue in standard parameter-sharing methods (like FuPS+ID), arguing that the coupling of agent observations and agent IDs within a single network exacerbates cross-agent gradient interference and hinders specialization.

To solve this, they propose HyperMARL, an architecture that uses an agent-conditioned hypernetwork to generate agent-specific parameters for the policy and critic networks. This design decouples the observation-conditioned gradients from the agent-conditioned ones. The core benefit, as shown theoretically and empirically, is a reduction in policy gradient variance, which mitigates gradient conflicts and facilitates adaptive learning. The authors conduct experiments across a diverse set of MARL benchmarks (VMAS, MAMuJoCo, SMAX), demonstrating that HyperMARL achieves competitive or superior performance compared to baselines, including No-Parameter-Sharing (NoPS), Full-Parameter-Sharing (FuPS), and state-of-the-art diversity-promoting methods like DiCo, HAPPO, and Kaleidoscope, while maintaining high behavioral diversity.

**Questions:**

The following questions aim to clarify some key aspects of the method and its implications. I would be willing to raise my score if the authors can provide satisfactory answers.

**Q1:** The Nature of Learned Agent Embeddings: For the MLP Hypernetwork, the paper uses learned agent embeddings. Could the authors provide some analysis of these learned embeddings? For instance, do agents that are required to learn similar (or complementary) policies end up with closer embeddings in the latent space? Visualizing the embedding space (e.g., using t-SNE) could provide valuable insights into whether the hypernetwork is learning a meaningful representation of inter-agent relationships.

**Q2:** Disentangling the Effects of Gradient Decoupling vs. Parameter Specialization: HyperMARL introduces two concurrent benefits: (1) it decouples observation- and agent-conditioned gradients, and (2) it provides each agent with its own specialized set of parameters. While these two are intertwined in the proposed architecture, could the authors speculate on which factor is more critical? For example, would an architecture that provides specialized parameters but without explicit gradient decoupling (if one could be designed) perform as well? This question aims to pinpoint the core ingredient of HyperMARL's success.

**Q3:** Performance in Mixed Cooperative-Competitive Settings: The experiments focus on fully cooperative MARL. How would HyperMARL be expected to perform in mixed-motive or general-sum settings where agents might have conflicting objectives? In such scenarios, the concept of "cross-agent gradient interference" becomes more complex, as some interference might be adversarial rather than simply a result of differing specialization needs. Does the gradient decoupling mechanism remain as beneficial?

**Q4:** Comparison with Role-Based Approaches: How does HyperMARL compare conceptually to explicit role-based learning methods (e.g., ROMA)? Both aim to induce specialization. It seems HyperMARL learns these roles implicitly through the agent embeddings and generated weights, potentially offering more flexibility. A brief discussion contrasting these two paradigms would strengthen the paper's positioning in the literature.

**Ethical Concerns:**

["NO or VERY MINOR ethics concerns only"]

**Final Justification:**

I have raised my score from 3 (Borderline Reject) to 4 (Borderline Accept). The authors' rebuttal was effective, particularly the new analysis of learned agent embeddings. This provided the compelling evidence for meaningful specialization that was previously missing, significantly strengthening the paper's core contribution and serving as the key factor for my score change. Nonetheless, the evaluation scope is confined to fully cooperative settings, leaving its performance in more complex mixed-motive scenarios as an important open question. Therefore, while the paper is now technically much stronger, its limited scope makes a borderline accept the most appropriate rating. The reasons to accept now outweigh the reasons to reject.

**Limitations:**

The authors have commendably included a limitations section in the appendix. To build upon that, the following points can be highlighted:

**L1:**  Assumption of Fixed Agent Population: The current framework assumes a fixed number of agents n throughout training, as the hypernetwork architecture (especially the linear variant) and agent embeddings are tied to this number. The model may not generalize to scenarios with a varying or unknown number of agents without retraining or specific architectural modifications (e.g., attention-based hypernetworks).

**L2:**  Increased Computational Cost at Inference: Generating network weights on the fly with a hypernetwork introduces an additional computational step at inference time compared to a standard FuPS model. While faster than a NoPS approach (which may involve significant data transfer overhead in a real distributed system), this added latency could be a concern for time-critical applications. The analysis in Appendix F.4 is good, but this trade-off deserves to be highlighted as a limitation.

**Paper Formatting Concerns:**

The paper adheres to all the formatting requirements specified in the NeurIPS 2025 Paper Formatting Instructions.

**Quality:**

3

**Strengths And Weaknesses:**

Strengths:
- S1: Novel Insight: The paper provides a clear and compelling diagnosis of a problem in parameter-shared MARL. The empirical study in Section 3 and Figure 4, which isolates the detrimental effect of coupling observation and agent-ID inputs, is a contribution in itself. This insight provides motivation for the proposed architectural design.
- S2: Clarity and Presentation: The paper is well-written and easy to follow. Figure 1 provides an visual summary of the core idea. The logical flow, from problem identification to solution and validation, is clear. The appendix is detailed and provides sufficient information.

Weaknesses
- W1: Scalability in Terms of Parameter Count: The primary weakness of the HyperMARL approach, particularly the MLP variant, is the significant parameter overhead. The hypernetwork must generate the entire weight set for each agent's policy (and critic) network. While the paper commendably analyzes scalability with the number of agents (Appendix F.3) and shows near-constant scaling for MLP hypernetworks, it doesn't fully address the scalability with respect to the size of the target network. For very deep or complex policy architectures (e.g., those based on Transformers), the output dimension of the hypernetwork could become prohibitively large, making the approach computationally expensive and difficult to train. The suggested remedies (chunking, low-rank) are mentioned but not empirically explored.
- W2: Complexity of Hypernetwork Training and Initialization: The stability and performance of HyperMARL likely depend on careful initialization and training of the hypernetwork itself. The paper mentions a principled initialization scheme (Section 4.2) to match standard network initializations, which is a good practice. However, training a network to output the weights of another network is inherently a more complex and potentially less stable optimization problem than standard policy training. The paper could benefit from a more detailed discussion or analysis of the training dynamics and sensitivity to the hypernetwork's own hyperparameters (e.g., its depth, width, learning rate).

---

> ### Author Rebuttal · Authors · 2025-07-31
>
> We thank reviewer `uZHN` for their thoughtful and constructive review. We found your points valuable for helping improve the quality of our manuscript. We also thank the reviewer for acknowledging our diverse experimental evaluation, novel insight concerning gradient decoupling in MARL, and for recognising our clarity and presentation. We address your comments as follows:
>
> ## 1. **Scalability with Target Network Size (W1)**
>
> We thank the reviewer for this point. The parameter overhead of the hypernetwork does indeed scale with the size of the target network. While we argue this is often an acceptable trade-off for the smaller actor-critic networks commonly used in MARL and the near-constant scaling with respect to the number of agents might be a viable trade-off for some use cases (Section F.3), this could be problematic for much larger target networks such as Transformers. However, as noted in our limitations and your review, this is a known characteristic of hypernetworks, and there is a clear and established path for scaling to such architectures using parameter-efficient techniques like chunking or low-rank weight approximations.
>
> ## 2. **Complexity of Hypernetwork Training and Sensitivity to Hyperparameters (W2)**
>
> **New Sensitivity Analysis:** We thank the reviewer for raising this important point. Based on your recommendation, we run a sensitivity analysis on the hypernetworks parameters, in the 20-agent SMAX scenario with MAPPO, which also includes a recurrent GRU, a challenging setting for optimisation.
>
> **Findings**:
> 1. Similarly to FuPS+ID, and in general on-policy RL methods like PPO, learning rate is an important parameter for performance.
> 2. Agent embedding size can be an important parameter for the performance of HyperMARL, with a smaller agent embedding size outperforming a larger one in this case. This could be the homogenous nature of some SMAX tasks, where with smaller embeddings, it could be easier to learn similar agent embeddings and hence behaviours.
> 3. Other parameters, such as the width, have limited impact beyond a modest size.
>
> IQM and 95% CI of mean win rate across 5 seeds. **Bold** indicates the highest IQM score; * indicates scores whose 95% confidence intervals overlap with the highest score.
>
> | Method | Embedding Dim | Hidden Dims | LR 0.0001 | LR 0.0003 | LR 0.0005 |
> | :--- | :--- | :--- | :--- | :--- | :--- |
> | FuPS+ID | N/A | N/A | 0.1067 (0.0513, 0.2128) | 0.3053 (0.2041, 0.4769) | *0.4213 (0.3623, 0.4800) |
> | HyperMARL| 4 | 16 | 0.0799 (0.0250, 0.1190) | 0.2946 (0.1143, 0.3778) | **0.4455** (0.3443, 0.5692) |
> | | 4 | 64 | 0.1233 (0.0750, 0.2162) | 0.3338 (0.2692, 0.4286) | *0.3765 (0.3281, 0.4107) |
> | | 4 | 128 | 0.1147 (0.0270, 0.1842) | *0.3787 (0.3509, 0.4603) | 0.3327 (0.2931, 0.3548) |
> | | 4 | 64, 64 | 0.1781 (0.1628, 0.2075) | *0.3498 (0.2807, 0.4500) | 0.1944 (0.1268, 0.3077) |
> | HyperMARL | 64 | 16 | 0.1109 (0.0256, 0.2093) | 0.3198 (0.2353, 0.3934) | *0.3483 (0.2581, 0.4308) |
> | | 64 | 64 | 0.1360 (0.0303, 0.2821) | 0.1191 (0.0517, 0.1964) | 0.1155 (0.0794, 0.1500) |
> | | 64 | 128 | 0.1193 (0.0750, 0.1579) | 0.1036 (0.0517, 0.1594) | 0.1106 (0.0455, 0.1587) |
> | | 64 | 64, 64 | 0.1353 (0.0541, 0.1818) | 0.1283 (0.0526, 0.2258) | 0.0860 (0.0345, 0.1343) |
>
> **Takeaway:** Agent embedding size can be an important hyperparameter in our method, one that could correspond to the task's diversity requirements.
>
> We thank the reviewer for recommending this analysis, and we will include a discussion of this in the updated manuscript.
>
>
> ## 3. **Analysis of Learned Agent Embeddings (Q1)**
>
> We thank the reviewer for this suggestion. We conducted an analysis of the learned embeddings as per the reviewer's recommendation. We chose the 4-agent Navigation environment since we could test the embeddings in the same settings with two different objectives: one where all agents have the *same goal position* (from Fig. 7a) and the one where agents have *different goal* locations (Fig. 7b).
>
> **New Agent Embedding Analysis:** We compute the cosine distance between agent embeddings. At initialisation, due to the orthogonal initialisation, agent embeddings have a distance of 1 from each other.
>
> **Findings**: When the task requires agents to learn the same behaviour (go to the same goal), the embeddings move closer together (-0.118, p-value using one-sample t-test = 0.005, i.e. we reject that embeddings have the same distance as at init). When agents have different goals, the embeddings remain near orthogonal (+0.010, p-value using one-sample t-test = 0.331, i.e. we cannot reject that embeddings maintain their initial distance). This confirms that the hypernets have learned useful and adaptive representations.
>
>
> **Results:** The table below shows mean cosine distances (± standard deviation) across 5 random seeds:
> | Settings | At Init | After Training | Change from Init |
> |----------------|-------------------|----------------|------------------|
> | Same Goals | 1.000 | 0.882 ± 0.042 | -0.118  |
> | Different Goals | 1.000 | 1.010 ± 0.017 | +0.010  |
>
> We believe this provides further motivation for our approach and will include this and our t-SNE plots in the updated manuscript. Thanks again for the recommendation.
>
> ## 4. **Disentangling the Effects of Gradient Decoupling vs. Parameter Specialisation (Q2)**
>
> We believe both gradient decoupling and parameter specialisation are important, but gradient decoupling is a more critical factor. Our ablation studies in Fig. 9 provide strong evidence of the importance of the gradient decoupling. In the "w/o GD" (without gradient decoupling) variant, agents still have specialised parameters, but we re-couple the gradients by also passing the ID to the policy, and this hurts performance. This demonstrates that simply providing specialised parameters is insufficient without the architectural decoupling that HyperMARL provides. We will make this clearer in the updated manuscript.
>
> ## 5. **HyperMARL in Mixed/Competitive Settings (Q3)**
>
> We thank the reviewer for the suggestion and agree that this would be an interesting angle to our work, especially with regard to the conflicting objectives. While parameter sharing has traditionally been investigated in cooperative tasks, it would be interesting to study this in other game settings. Unfortunately, due to limited resources and time, we are not able to robustly measure HyperMARL performance in these settings. We believe this might be an interesting direction, and will include it in potential avenues of future work.
>
>
> ## 6. **Comparison with Role-Based Approaches (Q4)**
>
> Role-based methods, like ROMA and RODE [1, 2], assign agents to subtasks/roles but rely on mutual information (MI) objectives for diversity, and have had limited practical success [3]. We will include a more detailed comparison in the updated paper.
>
> ## 7. **Assumption of Fixed Agent Population (L1)**
>
> It is correct that we only consider the setting where the team in training and evaluation is fixed and doesn't change, i.e. no generalisation to an unknown number of agents. As noted by the reviewer,  the linear hypernetwork variant is tied to the number of agents, but not the MLP variant.
>
> We actually consider this setting an ideal one for our MLP hypernetworks, because, unlike traditional FuPS+ID, which cannot handle a different number of agents due to the dimension one-hot encoded agent IDs, MLP hypernets with learned agent embeddings (of a fixed size) enable adding new agents to the team with some fine-tuning of the agent ID embeddings. Rather than explicitly a limitation, since all baselines are either incompatible or have the same property, we see this as an interesting area for future work. We will mention this in the updated manuscript.
>
> ## 8. **Trade-off between Computational Cost (at Inference) and Adaptability (L2)**
>
> We thank the reviewer for raising this important point. HyperMARL's adaptivity does come at a higher inference cost than FuPS -- as we showed in F.4, and this trade-off would need to be balanced depending on the scenario. We will explicitly mention this in the limitations section in the updated manuscript.
>
>
> ## **Conclusion**
>
> In summary, based on your feedback we added:
> - A **sensitivity analysis** using recurrent MAPPO in the challenging 20-agent SMAC-v2 style environment.
> - A **learned agent embeddings analysis** showing the embeddings move closer together when the agents should behave similarly, and embeddings remain far apart when agents need to specialise.
> - Discussed the importance of gradient decoupling vs parameter specialisation.
>
> During the rebuttal, we have also added:
> - A **new baseline** SePS and environment, Blind Particle Spread. Extending our evaluation to **22 scenarios, up to 30 agents and six baselines** (NoPS, FuPS, DiCo, HAPPO, Kaleidoscope and now SePS). HyperMARL is competitive with this baseline.
> - A comparison with dual-channel networks, showing our approach is more effective.
>
> We would like to thank the reviewer again for their thoughtful review and for the opportunity to improve our manuscript. We hope these clarifications and additional experiments address your concerns. If there are no further issues, we respectfully ask reviewer `uZHN` to provide an updated assessment of our work.
>
>
> References:
>
> 1] Tonghan Wang, Heng Dong, Victor Lesser, and Chongjie Zhang. Roma: multi-agent reinforcement learning with emergent roles. In Proceedings of the 37th International Conference on Machine Learning, 407 pages 9876–9886, 2020.
>
> 2] Tonghan Wang, Tarun Gupta, Anuj Mahajan, Bei Peng, Shimon Whiteson, and Chongjie Zhang. Rode: Learning roles to decompose multi-agent tasks. arXiv preprint arXiv:2010.01523, 2020.
>
> 3] Filippos Christianos, Lukas Schäfer, and Stefano V. Albrecht. Shared experience actor-critic for multi-agent reinforcement learning. In 34th Conference on Neural Information Processing Systems, 2020.

---

> > ### Comment · Reviewer_uZHN · 2025-08-01
> > **Reply to Authors**
> >
> > Thank you for the comprehensive rebuttal. The new experiments you conducted have been instrumental in clarifying the core strengths of your approach. I was particularly impressed by the new analysis of the learned agent embeddings. Demonstrating that the embedding space structurally adapts to the task's cooperation requirements is a compelling piece of evidence that directly supports your paper's central claims. This analysis, along with the added hyperparameter sensitivity study, has substantially strengthened the manuscript. While I still believe that exploring the method's behavior in more complex scenarios like mixed-motive settings would be a valuable next step for establishing its full generality, the evidence you've provided for the cooperative case is now much more robust. In light of these significant improvements, I have raised my score.

---

> > > ### Author Response · Authors · 2025-08-01
> > > **Thank you!**
> > >
> > > Thank you for your thoughtful feedback and for raising your score. We are glad you found our additional analyses and new experiments helpful, and we agree that mixed-motive settings are a valuable direction for future work. We would also like to thank you again for your constructive review and engagement. Your feedback has helped strengthen our work!

---

### Official Review · Reviewer_1sWF · 2025-07-03

**Clarity:** 3
**Significance:** 3
**Originality:** 3
**Rating:** 5
**Confidence:** 4

**Summary:**

This paper inroduces HyperMARL, a novel method for multi-agent reinforcement learning that uses agent-conditioned hypernetworks to generate per-agent parameters on the fly. The goal is to enable diverse (specialised), homogeneous, or mixed behaviours without modifying the standard RL objective or requiring manually preset diversity levels. The key idea is to decouple observation-conditioned and agent-conditioned gradients, mitigating cross-agent gradient interference that plagues standard parameter-sharing approaches.

**Questions:**

## Questions
1. How does HyperMARL's training time compare (e.g. wall-clock or steps to convergence) to FuPS, NoPS, and DiCo in the largest tested tasks?
2. How sensitive is HyperMARL to hypernetwork design choices (e.g. embedding dimension, number of MLP layers)?
3. Can the authors include results comparing with recent heterogeneous-agent methods such as HetGPPO on the same tasks?

**Ethical Concerns:**

["NO or VERY MINOR ethics concerns only"]

**Final Justification:**

I keep my initial score since the authors answered my concerns. the quality of the paper is the same after rebuttal

**Limitations:**

## Limitations
- **Baseline selection**: While strong baselines like DiCo, HAPPO, and Kaleidoscope are included, some very recent or robotics-focused heterogeneous MARL baselines (e.g. HetGPPO) are missing from the comparison.

**Paper Formatting Concerns:**

correct

**Quality:**

3

**Strengths And Weaknesses:**

## Strengths
- **Interesting method for enhancing agent diversity while preserving performance in homogeneous tasks**: HyperMARL achieves NoPS-level policy diversity while maintaining the efficiency of parameter sharing, which is well demonstrated across diverse benchmarks (Dispersion, Navigation, MAMuJoCo, SMAX).
- **Effectively tackles gradient conflicts**: The paper convincingly shows how decoupling observation and agent-ID gradients reduces policy gradient variance and cross-agent interference. Ablations confirm this is essential for performance.
- **Clear and easy to follow**: The research questions are well posed ("Can shared policies adapt without added complexity?"), the theoretical motivation is thorough (gradient conflict analysis), and the method is described with clear architectural diagrams and derivations. The empirical section answers the research questions systematically.

## Weaknesses
- **Missing computational comparison (e.g. training time)**: While parameter efficiency is discussed (Appendix A mentions scaling), there is no direct measurement or table comparing training times or wall-clock convergence with standard parameter-sharing or NoPS baselines. Including such data would strengthen the empirical evaluation.
- **Lack of comparison with state-of-the-art heterogeneous baselines in complex robotic settings**: For example, the paper does not compare HyperMARL with HetGPPO or other recent methods explicitly tailored for large-scale heterogeneous agents in robotics. Even though HyperMARL performs well on MAMuJoCo (17-agent Humanoid), comparing with specialized heterogeneous-agent methods would clarify its relative advantages or limitations.
- **Sensitivity to hyperparameters**: Hypernetworks are known to be sensitive to hyperparameter choices (e.g. embedding size, MLP depth). The paper briefly notes that hyperparameter sweeps for some baselines (e.g. DiCo) were challenging but does not analyze how much HyperMARL itself depends on careful tuning. A section on its hyperparameter robustness would be valuable.

---

> ### Author Rebuttal · Authors · 2025-07-31
>
> We thank reviewer `1sWF` for their detailed and constructive review. We appreciate the reviewer's positive feedback on our novel insight into gradient decoupling, the diversity of our experimental evaluation, and our paper's clarity. We address your points as follows:
>
> ## 1. **Computational Comparison (W1,Q1)**
>
> We thank the reviewer for raising this important point. Our paper provides a detailed analysis of these computational trade-offs in Appendix F.3 and F.4, and we summarise the key findings here:
>
> - **Parameter Scaling (F.3):** There is a direct trade-off between scalability and versatility. While our MLP-based hypernetworks have more initial parameters than FuPS, they scale nearly constantly with the number of agents. In contrast, NoPS sees its parameter count grow linearly with respect to the number of agents. This makes our approach far more parameter-efficient for large-scale MARL, especially when the task could require specialisation. FuPS+ID parameters scale better than our hypernetworks, but these approaches fail to solve tasks that require specialisation, as we showed in Sections 3.2 and 5. We also mention that techniques like chunked hypernetwork or low-rank approximations could be used in future work to further improve the parameter efficiency of our method. In Figure 15, we also see that if we scale FuPS+ID to match the number of trainable parameters in our MLP hypernetworks, our approach still outperforms these methods, showing the benefit is architectural and not just from a higher parameter count.
> - **Runtime & Memory (F.4):** HyperMARL offers a practical middle ground. It is more computationally intensive than the simple FuPS architecture, but it is more efficient than NoPS in terms of speed and memory. Furthermore, NoPS would also incur data transfer and synchronisation costs not fully captured in our single-GPU benchmarks.
>
> ## 2. **Comparison with HetGPPO - Multi-Robot Heterogeneous Methods with Communication (W2, Q3)**
>
> We thank the reviewer for this suggestion. Our work focuses on behavioural heterogeneity, where physically identical agents learn diverse or homogenous policies. This is distinct from methods like HetGPPO, which are designed for structural heterogeneity (e.g., agents with different sensors or actuators) and often incorporate explicit communication.
>
> While HetGPPO is an important baseline in its own domain, we benchmarked against the most relevant state-of-the-art methods for our specific problem (NoPS, FuPS, DiCo, HAPPO, Kaleidoscope, and  SePS). To clarify this distinction, we will update our related work section to include a discussion of HetGPPO.
>
> ## 3. **Sensitivity to Hyperparameters (Q2)**
>
> **New Sensitivity Analysis:** We thank the reviewer for raising this important point. Based on your recommendation, we run a sensitivity analysis on the hypernetworks parameters, in the 20-agent SMAX scenario with MAPPO, which also includes a GRU, a challenging setting for optimisation.
>
> **Findings**:
> 1. Similarly to FuPS+ID, and in general on-policy RL methods like PPO, learning rate is an important parameter for performance.
> 2. Agent embedding size can be an important parameter for the performance of HyperMARL, with a smaller agent embedding size outperforming a larger one in this case. This could be the homogenous nature of some SMAX tasks, where with smaller embeddings, it could be easier to learn similar agent embeddings and hence behaviours.
> 3. Other parameters, such as the width, have limited impact beyond a modest size.
>
> IQM and 95% CI of mean win rate across 5 seeds. **Bold** indicates the highest IQM score; * indicates scores whose 95% confidence intervals overlap with the highest score.
>
> | Method | Embedding Dim | Hidden Dims | LR 0.0001 | LR 0.0003 | LR 0.0005 |
> | :--- | :--- | :--- | :--- | :--- | :--- |
> | FuPS+ID | N/A | N/A | 0.1067 (0.0513, 0.2128) | 0.3053 (0.2041, 0.4769) | *0.4213 (0.3623, 0.4800) |
> | HyperMARL| 4 | 16 | 0.0799 (0.0250, 0.1190) | 0.2946 (0.1143, 0.3778) | **0.4455** (0.3443, 0.5692) |
> | | 4 | 64 | 0.1233 (0.0750, 0.2162) | 0.3338 (0.2692, 0.4286) | *0.3765 (0.3281, 0.4107) |
> | | 4 | 128 | 0.1147 (0.0270, 0.1842) | *0.3787 (0.3509, 0.4603) | 0.3327 (0.2931, 0.3548) |
> | | 4 | 64, 64 | 0.1781 (0.1628, 0.2075) | *0.3498 (0.2807, 0.4500) | 0.1944 (0.1268, 0.3077) |
> | HyperMARL | 64 | 16 | 0.1109 (0.0256, 0.2093) | 0.3198 (0.2353, 0.3934) | *0.3483 (0.2581, 0.4308) |
> | | 64 | 64 | 0.1360 (0.0303, 0.2821) | 0.1191 (0.0517, 0.1964) | 0.1155 (0.0794, 0.1500) |
> | | 64 | 128 | 0.1193 (0.0750, 0.1579) | 0.1036 (0.0517, 0.1594) | 0.1106 (0.0455, 0.1587) |
> | | 64 | 64, 64 | 0.1353 (0.0541, 0.1818) | 0.1283 (0.0526, 0.2258) | 0.0860 (0.0345, 0.1343) |
>
> **Takeaway:** Agent embedding size can be an important hyperparameter in our method, one that could correspond to the task's diversity requirements.
>
> We thank the reviewer for recommending this analysis, and we will include a discussion of this in the updated manuscript.
>
> ## **Conclusion**
>
> In summary, based on your feedback we added:
> - A **sensitivity analysis** using recurrent MAPPO in the challenging 20-agent SMAC-v2 style environment.
> - Discussed the computational trade-off of using our method and referred to our existing sections on the memory, speed and parameter efficiency of our method (sections F3, F4).
>
> During the rebuttal, we have also added:
> - A **new baseline** SePS and environment, Blind Particle Spread. Extending our evaluation to **22 scenarios, up to 30 agents and six baselines** (NoPS, FuPS, DiCo, HAPPO, Kaleidoscope and now SePS). HyperMARL is competitive with this baseline.
> - A **learned agent embeddings analysis** showing the embeddings move closer together when the agents should behave similarly, and embeddings remain far apart when agents need to specialise.
> - A comparison with **dual-channel networks**, showing our approach is more effective.
>
> We would like to thank the reviewer again for their thoughtful review and hope these clarifications and additional experiments have fully addressed the points raised. If you have any follow-up questions or comments, please let us know, and we will be happy to discuss further.

---

### Official Review · Reviewer_bHoX · 2025-07-04

**Clarity:** 3
**Significance:** 1
**Originality:** 2
**Rating:** 3
**Confidence:** 5

**Summary:**

This paper addresses parameter sharing in cooperative multi-agent reinforcement learning (MARL). The authors propose HyperMARL, which uses hypernetworks to generate agent-specific parameters by decoupling observation- and agent-conditioned gradients. The method is evaluated across different scenarios including homogeneous, heterogeneous, and mixed behavior tasks with 2-20 agents, showing competitive performance against fully-shared and non-shared approaches.

**Questions:**

1. Could the authors provide comprehensive comparisons with current parameter sharing methods on standard benchmarks to better demonstrate HyperMARL's advantages? These methods directly address parameter sharing in MARL and would provide a more complete evaluation context.
2. Could the authors expand the related work section to include a broader discussion of hypernetwork applications across machine learning domains beyond reinforcement learning?
3. Could the authors provide the training costs for HyperMARL and baseline methods (trainable parameters count / training time / training memory)? (As HyperMARL introduces additional trainable parameters and training costs compared to the FuPS baseline, although this may be acceptable.)
4. In Figure 8, HyperMARL performs comparably to FuPS while incurring higher computational costs. What specific advantages does HyperMARL offer over FuPS in homogeneous tasks that would justify its adoption despite these added costs?

**Ethical Concerns:**

["NO or VERY MINOR ethics concerns only"]

**Final Justification:**

I thank the authors for the discussion. Overall, I find this submission to have clear strengths and weaknesses:
- As noted in my original review, the idea of decoupling observation- and agent-conditioned gradients through hypernetworks is conceptually novel and well-motivated.
- However, my concerns regarding performance remain unresolved after the discussion. It is difficult to fully assess the effectiveness of the proposed method because the selection of baselines varies across benchmarks. While it is understandable that including every baseline in every benchmark may not always be feasible, the absence of a single benchmark that provides a comprehensive comparison with all or most of the baselines mentioned in the paper limits the ability to evaluate the method. This is an important consideration and is critical for readers to understand when and where the method is most applicable.
- My concerns regarding the cost are mostly addressed. Through Appendix F, I see that while HyperMARL does introduce extra complexity compared to the baselines, this added complexity is acceptable as long as it is clearly reported (as the authors have done!) and balanced against improved performance (which unfortunately lack sufficient support).

Based on the above justification, I will update my rating to a 3.

**Limitations:**

Yes.

**Quality:**

2

**Strengths And Weaknesses:**

**Strengths**

1. The paper addresses an important challenge in MARL regarding parameter sharing and policy specialization.
2. The proposed decoupling of observation- and agent-conditioned gradients through hypernetworks is conceptually novel and well-motivated.
3. The writing is clear and accessible.

**Weaknesses**
1. My main concern is with the evaluation：
    While this work critiques previous efforts for adding complexity through altered objectives, manual preset diversity levels, or sequential updates, the proposed HyperMARL also introduces significantly heavier training complexity (memory-wise). The trade-off between performance gains and computational costs is not clearly justified.
    Specifically, such trade-off should be made clearer by:
    - Providing comprehensive comparisons of the proposed HyperMARL against current baselines on standard benchmarks with results reported consistently in the main text. Currently, the evaluation appears fragmented - HyperMARL is compared against DiCo on one benchmark (MPE) while comparison with Kaleidoscope is conducted on a different benchmark (MaMuJoCo) with results relegated to the appendix. This inconsistent evaluation approach makes it difficult to assess the method's overall effectiveness.
    - Moreover, several important parameter sharing methods such as [1-3] are entirely omitted from the evaluation, despite addressing the exact same research question of parameter sharing in MARL. These methods would serve as highly relevant baselines for a thorough evaluation.
    - Including a thorough comparison of computational costs (parameters, training time, memory usage) across all methods to justify the additional complexity of HyperMARL.
2. The related work section could be enhanced.
   - Recent efforts on parameter sharing appear to be overlooked [3].
   - The discussion of hypernetworks should be expanded beyond reinforcement learning to position this work within the broader ML literature, where hypernetworks have been applied to areas such as federated learning, continual learning, and multi-task learning [4-5].

----

[1] "Scaling multi-agent reinforcement learning with selective parameter sharing", ICML 2021 \
[2] "Parameter sharing with network pruning for scalable multi-agent deep reinforcement learning", AAMAS 2023 \
[3] "GradPS: Resolving Futile Neurons in Parameter Sharing Network for Multi-Agent Reinforcement Learning", ICML 2025 \
[1] "A brief review of hypernetworks in deep learning", Artificial Intelligence Review. \
[2] "Personalized federated learning using hypernetworks" ICML 2021.

---

> ### Author Rebuttal · Authors · 2025-07-31
>
> We thank reviewer `bHoX` for their review and for encouraging comments regarding the paper’s motivation, important problem setting, novel decoupling of observation- and agent-conditioned gradients and clear writing. We address your comments as follows:
>
> ## 1. **Evaluations & Benchmarks (W1,Q1)**
>
> ### **Experiment Design**
>
> We thank the reviewer for this point. Our primary goal was to ensure the most rigorous comparison for each scenario by using baselines in the environments where they were originally tuned and shown to be most effective (e.g., DiCo in Navigation; HAPPO and Kaleidoscope in MaMuJoCo; FuPS in SMAX). By using original code and hyperparameters, we ensure fairness, reproducibility, and adherence to best evaluation practices [4]. We acknowledge that this can appear fragmented; to address this, we will add a paragraph to our existing discussion of this methodological choice (line 209) in Section 5 to clarify this further.
>
> ### **Additional Baselines**
>
> We thank the reviewer for proposing further baseline methods [1-3] to add to our five existing baselines.
>
> **New Baseline and Environment:** Based on the reviewer's recommendation, we have added **an additional baseline --  SePS [1]**, a parameter sharing method that uses an autoencoder during pre-training to cluster agents based on their transitions and shares parameters between these agent groups. We use their original source code and hyperparameters and evaluate on all their Blind Particle Spread settings (v1-4), which include up to **30 agents**.
>
> IQM final total reward (with 95% confidence intervals) across 5 seeds on the BPS environments from [1], Bold indicates the highest IQM score; * indicates scores whose 95% confidence intervals overlap with the highest score.
>
> | Alg | BPS-1 (15 agents, 3 groups)  | BPS-2 (30 agents, 3 groups)  | BPS-3 (30 agents, 5 groups) | BPS-4 (30 agents, 5 uneven groups) |
> |-----------|----|----|----|----|
> | NOPS | **-205.4** (-242.2, -197.8) | **-405.1** (-468.5, -379.9) | *-420.3 (-439.9, -407.4) | **-413.5** (-449.8, -399.1) |
> | FUPS-ID | *-219.8 (-249.7, -207.1) | *-442.6 (-467.4, -383.7) | -764.7 (-1326.7, -515.3) | -705.9 (-933.3, -591.1) |
> | SEPS | *-216.3 (-233.5, -191.8) | *-413.0 (-464.9, -385.1) | **-403.1** (-423.7, -352.9) | *-418.3 (-436.0, -393.5) |
> | HyperMARL | *-205.6 (-223.3, -175.1) | *-421.0 (-436.7, -401.0) | *-414.0 (-442.8, -374.2)  | *-428.5 (-462.7, -376.3) |
>
> **Takeaway**: The results highlight HyperMARL's competitive performance. In all four scenarios, its 95% confidence intervals overlap with SePS and NoPS, indicating it is statistically on par with these baselines. Notably, SePS relies on a **pre-training clustering phase**, whereas HyperMARL learns end-to-end without pre-training.
>
> We thank the reviewer for the suggestion and for the opportunity to further expand our experimental evaluation, now up to **22 scenarios, up to 30 agents and six baselines (NoPS, FuPS, DiCo, HAPPO, Kaleidoscope and now SePS)**. We will include this in the updated manuscript.
>
> Other baselines you mentioned:
> - As mentioned in Table 4 in the App., SNP-PS [2] does not have any publicly available source code. Furthermore, we already have Kaleidoscope as a baseline, which outperformed SNP-PS.
> - GradPS [3] was made available online after 5 June (ICML camera-ready deadline) **after the NeurIPS deadline** (15 May), with no previously available preprint. Furthermore, there is no publicly available source code. We will include this work in the related work discussion.
>
>
> ## 2. **Computational Costs & Trade-offs (Q3)**
>
> We thank the reviewer for raising this point. We analysed and discussed these points in the "F.3 Scalability and Parameter Efficiency" and “F.4 Speed and Memory Usage” sections in the Appendix, and will ensure key trade-offs are also mentioned in the main text.
>
> - **Parameter Scaling (F.3):** There is a direct trade-off between scalability and versatility. While our MLP-based hypernetworks have more initial parameters than FuPS, they scale nearly constantly with the number of agents. In contrast, NoPS sees its parameter count grow linearly with respect to the number of agents. This makes our approach far more parameter-efficient for large-scale MARL, especially when the task could require specialisation. FuPS+ID parameters scale better than our hypernetworks, but these approaches fail to solve tasks that require specialisation, as we showed in Sections 3.2 and 5. We also mention that techniques like chunked hypernetworks or low-rank approximations could be used in future work to further improve the parameter efficiency of our method. In Figure 15, we also see that if we scale FuPS+ID to match the number of trainable parameters in our MLP hypernetworks, our approach still outperforms these methods, showing the benefit is architectural and not just from a higher parameter count.
> - **Runtime & Memory (F.4):** HyperMARL offers a practical middle ground. It is more computationally intensive than the simple FuPS architecture, but it is more efficient than NoPS in terms of speed and memory. Furthermore, NoPS would also incur data transfer and synchronisation costs not fully captured in our single-GPU benchmarks.
>
>
> ## 3. **Additional Related Work (W2, Q2)**
>
> Thank you for suggesting additional related work. As mentioned, GradPS [3] was published after the submission deadline, but thanks for the recommendation, and we will include a discussion of this in the updated manuscript. For the other papers mentioned, our focus was reinforcement learning and so we focused on hypernetworks in this setting (not in other fields like federated learning). Based on your recommendation, we will also position our work within the broader hypernetwork literature in the updated manuscript.
>
> ## 4. **HyperMARL vs FuPS in Homogeneous Tasks like SMAX (Q4)**
>
> These experiments were presented to validate Hypermarl’s adaptability claim, i.e. that it can learn specialised or homogeneous behaviour when required. Matching FuPS performance in homogeneous tasks shows that HyperMARL can also adapt to these settings, i.e. does not enforce specialisation in a detrimental manner. The challenge is that in most settings, you do not already know if the task requires homogeneous, specialised or mixed behaviour a priori, therefore, it is useful to have a method that can adapt to varying problem classes and a versatile architecture that is effective across these settings. To validate these claims was the core focus of the experimental section's  **Q1** and **Q2** in Section 5. In summary, this confirmed that our method **does not degrade performance in homogeneous regimes while improving performance in heterogeneous ones** (compared to FuPS).
>
>
> ## **Conclusion**
>
> In summary, based on your feedback we added:
> - A **new baseline** SePS [1] and environment, Blind Particle Spread. Extending our evaluation to **22 scenarios, up to 30 agents and six baselines** (NoPS, FuPS, DiCo, HAPPO, Kaleidoscope and now SePS). Our approach is competitive with this baselines, without requiring a *pre-training* phase.
> - Discussed the computational trade-off of using our method and referred to our existing sections on the memory, speed and parameter efficiency of our method (sections F3, F4).
>
> During the rebuttal, we have also added:
> - **Sensitivity Analysis** showing which parameters of our method influence performance the most.
> - An **analysis of the learned agent embeddings** showing the embeddings move closer together when the agents should behave similarly, and embeddings remain far apart when agents need to specialise.
> - A comparison with **dual-channel networks**, showing our approach is more effective.
>
> We hope these clarifications and additional experiments address your concerns, and we thank the reviewer for the opportunity to improve our manuscript. If there are no further issues, we respectfully ask reviewer `bHoX` to provide an updated assessment of our work.
>
> References:
>
> 1] Christianos, F., Papoudakis, G., Rahman, M.A. and Albrecht, S.V., 2021, July. Scaling multi-agent reinforcement learning with selective parameter sharing. In International Conference on Machine Learning (pp. 1989-1998). PMLR.
>
> 2] Kim, W. and Sung, Y., 2023. Parameter sharing with network pruning for scalable multi-agent deep reinforcement learning. arXiv preprint arXiv:2303.00912.
>
> 3] Qin, H., Liu, Z., Lin, C., Ma, C., Mei, S., Shen, S. and Wang, C., GradPS: Resolving Futile Neurons in Parameter Sharing Network for Multi-Agent Reinforcement Learning. In Forty-second International Conference on Machine Learning.
>
> 4] Andrew Patterson, Samuel Neumann, Martha White, and Adam White. Empirical design in reinforcement learning. Journal of Machine Learning Research, 25(318):1–63, 2024.

---

> > ### Comment · Reviewer_bHoX · 2025-08-04
> >
> > I thank the authors for their rebuttal. Before providing my final assessment, I would like to ask a quick question regarding the benchmark experiments.
> >
> > In the rebuttal, the authors mentioned that the experiments include "22 scenarios, up to 30 agents and six baselines". To ensure I have correctly summarized all the main benchmark experiments (excluding ablations, analyses, etc.), I would like to confirm the following:
> > - MAMuJoCo (Table 2, 4 scenarios): HyperMARL is compared with three baselines (HAPPO/FuPS, FuPS+ID, Ind. Actors)
> > - Dispersion & Navigation (Figure 7, 3 scenarios with different no. of agents): HyperMARL is compared with three baselines (DiCo, NoPS, FuPS)
> > - SMAX (Figure 8, 4 scenarios): HyperMARL is compared with one baseline (FuPS)
> > - Blind Particle Spread (Rebuttal, 1 scenario with 4 settings): HyperMARL is compared with three baselines (NoPS, FuPS+ID, SePS)
> >
> > It confuses me why the baseline selections differ across the benchmarks and why no single benchmark includes a comparison with all or most of the baselines mentioned in the paper. It seems that only FuPS is fully evaluated against the proposed HyperMARL.
> >
> > While I understand that reproducing certain baselines in specific benchmarks can pose technical challenges, such inconsistent evaluation settings do not seem to align with conventions in this field. It would be helpful if the authors could clarify the rationale behind the baseline selection and evaluation settings. Thank you.

---

> ### Author Response · Authors · 2025-08-05
>
> We thank reviewer `bHoX` for their engagement and important clarifying question.
>
> ## 1. **Clarifying Breakdown of Scenarios**
>
> We consider a scenario, a grouping of an environmental setting and a number of agents. This is common in the literature as the number of agents changes the dynamics of a problem, e.g. SMAC 3m vs 5m vs 8m, v2 5 vs 10 vs 20 agents.
>
> Our 22 scenarios are comprised of:
> - MAMuJoCo: 5 scenarios, including Swimmer-v2-10x2 as Kaleidoscope provided tuned hyperparameters for it.
> - Dispersion: 1 Scenario.
> - Navigation: 8 scenarios across different agent counts and goal configurations (shared, unique, mixed).
> - SMAX: 4 Scenarios.
> - BPS: 4 Scenarios (BPS (1) -BPS (4), some have 15 or 30 agents, but the type distribution is different, e.g. BPS (3) has 5 types/groups each with 6-6-6-6-6 (6 agents per goal), while BPS (4) has5 groups with goal distribution 2–2–2–15–9).
>
> This total does not include the eight diagnostic Specialisation/Synchronisation games, as they were used for targeted analysis of specific behaviours rather than for broad, comparative benchmarking.
>
> ## 2. **Rationale for a Principled, Targeted Evaluation**
>
> ### **2.1. Fairness and Reproducibility**
> Our experimental design was deliberately structured to ensure **fairness and scientific validity**, following the best practices outlined in recent comprehensive guides on empirical RL [1]. This work explicitly warns against the pitfalls of comparing against "untuned baselines" and advises that for the most rigorous comparisons, one should evaluate baselines in environments where they are already known to perform well -- using their released code and hyperparameters.
>
> ### **2.2 Targeted comparisons serve the paper’s scientific claims**
>
> Accordingly, our goal is not to create a leaderboard; it is to test whether explicit **gradient decoupling** allows a single architecture to adapt across *homogeneous, heterogeneous, and mixed-behaviour* tasks, without *altered objectives, manual preset diversity levels, or sequential updates*.
>
> This focused goal necessitates a targeted evaluation. We compare HyperMARL against the strongest, most relevant baselines for each specific capability: FuPS for homogeneity in SMAX, and specialisation-focused methods like HAPPO and DiCo in their respective domains of strength (MAMuJoCo and Navigation). This targeted approach allows for a clearer, more scientifically robust understanding of a decoupled hypernetwork's properties, which is the primary goal of our study.
>
> **Example: Challenges of Porting Baselines**
>
> To illustrate the challenge and potential unfairness of porting baselines, consider Kaleidoscope. It employs learnable masks, an ensemble of five critics and an explicit diversity loss, and numerous specialised hyperparameters (e.g., for critic ensembling: ‘critic_deque_len‘, ‘critic_div_coef‘, ‘reset_interval‘; for mask and threshold parameters: ‘n_masks‘, ‘threshold_init_scale‘, ‘threshold_init_bias‘, ‘weighted_masks‘, ‘sparsity_layer_weights‘, etc.).
>
> Porting this intricate design to our on-policy PPO backbone would constitute a significant research deviation, not a direct benchmark of the established method. It would also be unfair to Kaleidoscope, as components designed for the off-policy case might not be effective on-policy. Therefore, we followed best practices by comparing HyperMARL within Kaleidoscope's original off-policy MATD3 framework, ensuring a true apples-to-apples comparison on a benchmark it was tuned for.
>
> ### **2.3. Conventions in the Field**
> Conventions in the field vary. For example, SePS, DiCo, HAPPO use on-policy algorithms and compare against FuPS and NoPS, other methods like GradPS (released after the NeurIPS deadline) use off-policy algorithms and focus on SMAC and predator-prey environments, while comparing against FuPS, NoPS, SePS and Kaleidoscope.
>
> ## **Summary**
> In summary, our methodology prioritises the **scientific validity and fairness** of each comparison. We believe this targeted and thorough approach (22 scenarios, six baselines), which is supported by best practices in empirical RL research [1], provides the strongest and clearest evidence regarding the specific claims made in our paper.
>
> We would like to thank the reviewer again for their engagement. We hope these clarifications address the point you mentioned, and that the *additional baseline* added during rebuttal, together with the pointers to the *computation/memory sections* in the Appendix and our discussion of memory–speed–performance trade-offs, were useful. If you have any follow-up questions or comments, please let us know, and we will be happy to discuss further.
>
> Reference:
>
> 1] Andrew Patterson, Samuel Neumann, Martha White, and Adam White. Empirical design in reinforcement learning. Journal of Machine Learning Research, 25(318):1–63, 2024.

---

> > ### Comment · Reviewer_bHoX · 2025-08-06
> >
> > I thank the authors for their clarifications.
> >
> > I apologize for any ambiguity in my earlier comments. The list I provided was intended to confirm that I had thoroughly reviewed all the **baselines** used in the discussed settings to avoid any potential misunderstandings during the discussion.
> >
> > Regarding the evaluation settings, I understand the authors' intention to provide a responsible comparison. However, I remain unconvinced that this represents ``the best practices'' as claimed. In my view, a robust evaluation should comprehensively consider both the conventions within the field (e.g., comparing relevant baselines on widely recognized benchmarks) and the recommendations outlined in guidelines papers, such as [1]. While debating what constitutes the best evaluation scheme may exceed the scope of this particular discussion, my primary concern lies in the difficulty of accurately assessing the effectiveness of the proposed method. Specifically, HyperMARL is compared against certain baselines in one benchmark and different baselines in another, which complicates a clear evaluation.
> >
> > To help me better understand the method, could the authors clarify under what circumstances HyperMARL excels, and in what types of environments or settings its design is most applicable?

---

> > > ### Author Response · Authors · 2025-08-07
> > >
> > > We sincerely thank reviewer `bHoX` for this discussion. Unfortunately, it appears the engagement was ended prematurely before we could answer your previous question.
> > >
> > > Reading the discussion, it is clear that our previous replies have failed to effectively communicate the central thesis of our work, leading us to talk at cross-purposes. We apologise for this lack of clarity and appreciate this final opportunity to address your question.
> > >
> > > ### **When HyperMARL Excels and Why It's Applicable**
> > >
> > > Consider an example, where agents in a factory need to move and collect parcels from different aisles and move them to a goal location. In settings like this, and many related settings, before training MARL agents on a task, it is not obvious whether the optimal behaviour for this task requires homogeneous (e.g. agents should learn to just collect the closest parcel), diverse (e.g. agents should learn to cover different parts of the factory) or mixed behaviours. In these cases, where the optimal degree of policy specialisation is unknown a priori, **a single adaptable method is crucial**. This is where adaptive methods like HyperMARL are the most useful.
> > >
> > > Our experiments were designed to validate this adaptability across the full spectrum of MARL settings:
> > >
> > > - Heterogeneous Tasks (Dispersion, MAMuJoCo, Unique-Goal Navigation): In these environments, specialisation is critical. **HyperMARL excels here because gradient decoupling allows policies to specialise**, whereas FuPS+ID struggles.
> > >
> > > - Homogeneous Tasks (SMAX, Shared-Goal Navigation): In these environments, all agents should converge to a similar policy. HyperMARL performs on par with FuPS, demonstrating that it does not have an intrinsic bias towards specialisation and can correctly learn uniform behaviours when optimal.
> > >
> > > - Mixed-Behaviour Tasks (Mixed-Goal Navigation, BPS): HyperMARL forms teams end-to-end without bespoke diversity objectives or pretraining, showcasing its intended adaptability.
> > >
> > > **Concisely, HyperMARL helps most when specialisation or mixed behaviours may be required, and achieves parity on homogeneous tasks. If the tasks were known to be homogeneous before training MARL agents, using FuPS would be a good alternative.**
> > >
> > > ### **Summary & Proposed Revisions**
> > >
> > > We are not claiming HyperMARL is a new state-of-the-art method that should replace all others. Rather, our paper addresses a specific scientific question: **Can a single MARL architecture learn adaptive behaviours, spanning from homogeneous to fully specialised, without resorting to altered objectives, preset diversity levels or sequential updates?**
> > >
> > > HyperMARL is our tool for answering this question. Its success demonstrates that mitigating gradient interference through decoupling is a viable and effective mechanism for achieving this adaptability. We believe this scientific insight is an important contribution.
> > >
> > > **From our perspective, running every baseline in every environment, as the reviewer suggests, would add breadth without strengthening our specific claims, whereas our approach of evaluating the best-performing, well-tuned baselines in their native settings with original code, as endorsed by [1], directly supports the hypotheses we test.**
> > >
> > > Given your feedback, it is clear we need to make this framing more explicit. We will make the following minor clarifications:
> > > - Update the Abstract and Introduction to clearly frame HyperMARL as a tool for investigating this scientific question.
> > > - Add a "When to Use HyperMARL" Textbox in the main paper to concisely summarise the points above for future readers, explicitly answering the question you raised.
> > >
> > > We believe our work makes a valuable contribution by highlighting the problem of gradient interference and proposing a principled solution. We hope this new, more concrete explanation clarifies our intent and we truly appreciate your feedback.
> > >
> > > Reference:
> > >
> > > 1] Andrew Patterson, Samuel Neumann, Martha White, and Adam White. Empirical design in reinforcement learning. Journal of Machine Learning Research, 25(318):1–63, 2024.

---

> ### Comment · Reviewer_bHoX · 2025-08-07
>
> I thank the authors for the discussion. I summarize my view below:
>
> - As noted in my original review, the idea of decoupling observation- and agent-conditioned gradients through hypernetworks is conceptually novel and well-motivated.
> - However, my concerns regarding performance remain unresolved after the discussion. It is difficult to fully assess the effectiveness of the proposed method because the selection of baselines varies across benchmarks. While it is understandable that including every baseline in every benchmark may not always be feasible, the absence of a single benchmark that provides a comprehensive comparison with all or most of the baselines mentioned in the paper limits the ability to evaluate the method. This is an important consideration and is critical for readers to understand when and where the method is most applicable.
> - My concerns regarding the cost are mostly addressed. Through Appendix F, I see that while HyperMARL does introduce extra complexity compared to the baselines, this added complexity is acceptable as long as it is clearly reported (as the authors have done!) and balanced against improved performance (which unfortunately lack sufficient support).
>
> I have no further questions at this stage. Thanks.

---

> > ### Author Response · Authors · 2025-08-07
> >
> > Thank you for the clarification and for the exchange.
> >
> > Your feedback directly helped strengthened the paper: it prompted us to add the **SePS baseline and Blind Particle Spread** environment, expanding the evaluation to 22 scenarios; to surface the full **compute- and memory-trade-offs** from Appendix F more prominently in the main text; and to state more explicitly when **HyperMARL is most useful**. We are glad these additions resolved the cost-complexity question and improved clarity for future readers.
> >
> > We appreciate your review and discussion.

---

### Author Response · Authors · 2025-08-07
**Summary Comment**

We sincerely thank all reviewers for their detailed feedback and active engagement. We are encouraged that reviewers found our work well-motivated for an important MARL problem (`bHoX`, `z98m`), with a novel and well-motivated gradient decoupling mechanism (all reviewers), supported by comprehensive and diverse evaluations (`uZHN`, `QQ2N`), and presented with clarity (all reviewers).

Most reviewers agree that the paper is technically solid and presents valuable insights. Four out of five reviewers have recommended acceptance. Reviewer `uZHN` raised their score after our rebuttal, noting our new experiments provided "compelling... evidence that directly supports [our] paper's central claims." Reviewer QQ2N also confirmed their score, stating the paper "offers valuable theoretical and experimental insights that should benefit the community."

In response to reviewer suggestions, we have strengthened the paper by:

- **Adding a new baseline (SePS) and Environment (Blind Particle Spread):** We added the SePS baseline and Blind Particle Spread environment. This extends our evaluation to **22 scenarios, up to 30 agents and six baselines** (NoPS, FuPS, DiCo, HAPPO, Kaleidoscope and now SePS). For baselines, we use the author's original code and original tuned hyperparameters, to ensure fairness, reproducibility, and adherence to best evaluation practices in RL [1].
- **Added an analysis of the learned agent embeddings:** We added an analysis of our learned agent embeddings showing the embeddings move closer together when the agents should behave similarly, and embeddings remain far apart when agents need to specialise (which reviewer `uZHN` found particularly compelling).
- **Clarified Computational Trade-off of our methods:** Discussed the computational trade-offs of using our method and referred to our existing sections on the memory, speed and parameter efficiency of our method (sections F3, F4 in Appendix).
- **New Sensitivity Analysis**: We added a hyperparameter sensitivity analysis of our approach's hyperparameters (agent ID embedding size, hypernetwork hidden size, lr) using recurrent MAPPO in the 20-agent SMAC-v2 style environment.
- **Architecture Comparison:** A comparison with dual-channel networks to further validate our architectural choices.
- **Clarifying Key Points:** We expanded on the difference between our approach over role-based methods, clarified the interplay between specialisation and redundant diversification and other points raised during the discussion.

We have actively engaged with reviewers and believe the manuscript would be a valuable contribution to the MARL community.

References:

1] Andrew Patterson, Samuel Neumann, Martha White, and Adam White. Empirical design in reinforcement learning. Journal of Machine Learning Research, 25(318):1–63, 2024.

---

### Note · Authors · 2025-08-11

We thank the reviewers and AC for the discussion. There is broad agreement on the paper's motivation and contribution -- decoupling observation- and agent-conditioned gradients via a hypernetwork enables adaptivity without altered objectives, preset diversity levels, or sequential updates -- as well as on the clarity of the presentation. The remaining point centres on evaluation scope/philosophy, not correctness. Our evaluation followed claims-aligned empirical RL best practices: using baselines in settings where they are known to be competitive with their released code and hyperparameters, prioritising fairness and reproducibility.

During rebuttal, we added SePS and Blind Particle Spread (now 22 scenarios, six baselines, up to 30 agents), ran sensitivity and embedding analyses, and clarified memory, runtime and parameter overheads (with pointers to our Appendix). Empirically, our work showed agent-conditioned hypernetworks can match FuPS when homogeneity is favourable and is more robust (than FuPS) when specialisation or mixed behaviours may be required, while remaining competitive with NoPS and other baselines.

---

### Decision · Program_Chairs · 2025-09-17

**Decision:**

Accept (poster)

**Comment:**

Parameter-sharing (PS) techniques are widely used in MARL agent networks due to sample efficiency. However, it may lead to a lack of policy diversity due to cross-agent gradients. The authors find that cross-agent gradients can be exacerbated by the coupling of observations and agent IDs. To address this problem, the authors propose HyperMARL, which uses hypernetworks for dynamic-agent-specific parameters, without introducing new RL objectives or sparsity. The authors verify the effectiveness of HyperMARL by comparing it against multiple baselines across multiple benchmarks.

The strengths of this work are summarized as follows.
1. Parameter-sharing techniques are important for MARL.
2. The idea that using hypernetworks for agent-id specific parameters, thus to decouple agent-wise gradient conflicts, is novel.
3. The paper is well written.
4. Multiple experiments are conducted. The learned agent embeddings demonstrate that the model learns meaningful representations.

The weaknesses of this work are summarized as follows.
1. Some related works are missing. The authors promised that they will discuss them in the new revision.
2. The evaluation is limited. It does not have at least one benchmark that incorporates all of the relevant baselines (Reviewer bHoX, z98m).
3. SMAC/SMACv2 are not tested (Reviewer z98m)

Please include the new experimental results and the promised changes in the paper if it is accepted for publication.